# Mechanistically informed machine learning links non-canonical TCA cycle activity to Warburg metabolism and hallmarks of malignancy

Lihao Lin[1], Francesco Lapi[1], Bruno Giovanni Galuzzi[2,3,4], Marco Vanoni[1,2], Lilia Alberghina[1,2], Chiara Damiani[1,2]*

**1** Department of Biotechnology and Biosciences, University of Milano-Bicocca, Milan, Italy, **2** SYSBIO Centre of Systems Biology/ ISBE.IT, Milan, Italy, **3** Institute of Bioimaging and Complex Biological Systems, Segrate, Italy, **4** National Biodiversity Future Center, Palermo, Italy

* chiara.damiani@unimib.it

## Abstract

Cancer cells undergo extensive metabolic rewiring to support growth, survival, and phenotypic plasticity. A non-canonical variant of the tricarboxylic acid (TCA) cycle, characterized by mitochondrial-to-cytosolic citrate export, has emerged as critical for embryonic stem cell differentiation. However, its role in cancer remains poorly understood.

Here, we present a two-step computational framework to systematically analyze the activity of this non-canonical TCA cycle across over 500 cancer cell lines and investigate its role in shaping hallmarks of malignancy. First, we applied constraint-based modeling to infer cycle activity, defining two complementary metrics: *Cycle Propensity*, measuring the likelihood of its engagement in each cell line, and *Cycle Flux Intensity*, quantifying average flux through the reaction identified as rate-limiting. We identified distinct tumor-specific patterns of pathway utilization. Notably, cells with high *Cycle Propensity* preferentially reroute cytosolic citrate via aconitase 1 (ACO1) and isocitrate dehydrogenase 1 (IDH1), promoting $\alpha$-ketoglutarate ($\alpha$KG) and NADPH production. Elevated engagement of this cycle strongly correlated with Warburg-like metabolic shifts, including decreased oxygen consumption and increased lactate secretion.

In the second step, to uncover non-metabolic transcriptional signatures associated with non-canonical TCA cycle activity, we performed machine learning–based feature selection using ElasticNet and XGBoost, identifying robust gene signatures predictive of cycle activity. Over-representation analysis revealed enrichment in genes involved in metastatic behavior, angiogenesis, stemness, and key oncogenic signaling. SHapley Additive exPlanations (SHAP) further prioritized genes with the strongest predictive contributions, highlighting candidates for experimental validation. Correlation analysis of DepMap gene-dependency profiles revealed distinct vulnerability patterns

**Data availability statement:** The transcriptomic data and gene dependency scores used in this study are publicly available from the Cancer Cell Line Encyclopedia and DepMap. All code used for data preprocessing, flux sampling, and the machine learning pipeline—along with processed datasets and a reproducible workflow—is available at:

https://github.com/CompBtBs/noncanonical-tca-cancer-analysis.

**Funding:** This work was supported by funding from the European Union – NextGenerationEU under the PRIN 2022 PNRR call (CUP H53D23007680001) awarded to C.D., and by grants to the ISBE-SYSBIO infrastructure awarded to L.A. and M.V., as well as the "ELIXIRxNextGenerationIT" initiative (Code IR0000010 – CUP B53C22001800006) awarded to M.V. The funders had no role in study design, data collection and analysis, decision to publish, or preparation of the manuscript.

**Competing interests:** The authors have declared that no competing interests exist.

associated with non-canonical TCA cycle activity, outlining a characteristic landscape of genetic dependencies.

Together, our integrative framework uniting constraint-based metabolic modeling and machine learning systematically reveals how non-canonical TCA cycle dynamics underpin metabolic plasticity and promote malignant traits.

## Author summary

A non-canonical variant of the TCA cycle that bypasses several steps of the mitochondrial TCA cycle has recently been identified. This cycle involves exporting mitochondrial citrate to the cytoplasm, converting it to oxaloacetate and then to malate, with the import of malate back into the mitochondria completing the cycle. This non-canonical mitochondrial/cytosolic cycle has been linked to changes in stem cell identity. However, its functional role in tumor metabolism remains poorly understood. To explore this potential connection, we developed a computational framework that combines mechanistic metabolic modeling with machine learning to analyze the activity of this pathway across more than 500 cancer cell lines. Using this approach, we measured the activity of the non-canonical TCA cycle and identified gene expression programs that predict its activation. We found that cells utilizing this pathway exhibit a Warburg-like metabolic profile. The gene expression programs associated with this metabolic state are enriched in processes related to core hallmarks of malignancy, including metastatic behavior, angiogenesis, stemness, and key oncogenic signaling. Overall, our results demonstrate that this recently discovered pathway links metabolic rewiring with transcriptional programs that drive tumor aggressiveness and progression, suggesting a mechanistic connection between the activation of the non-canonical TCA cycle and the transcriptional states associated with malignancy.

## 1 Introduction

A non-canonical variant of the TCA cycle was recently described by Arnold-Jackson et al. [1] as a cross-compartment metabolic pathway that links mitochondrial and cytosolic metabolism, hereinafter referred to as the Cit-Mal Cycle (Fig 1A). Unlike the classical TCA cycle confined to mitochondria, the Cit-Mal Cycle involves the export of citrate to the cytosol via the SLC25A1 transporter, its cleavage by ATP citrate lyase (ACLY) into acetyl-CoA and oxaloacetate, and the subsequent reduction of oxaloacetate to malate by cytosolic malate dehydrogenase (MDH1). Engagement of this cycle has been reported in both normal and cancer cells and has been linked to transitions in cellular identity [1]. In embryonic stem cells, for instance, it is required to exit from naïve pluripotency, and its inhibition prevents proper differentiation.

PLOS Computational Biology

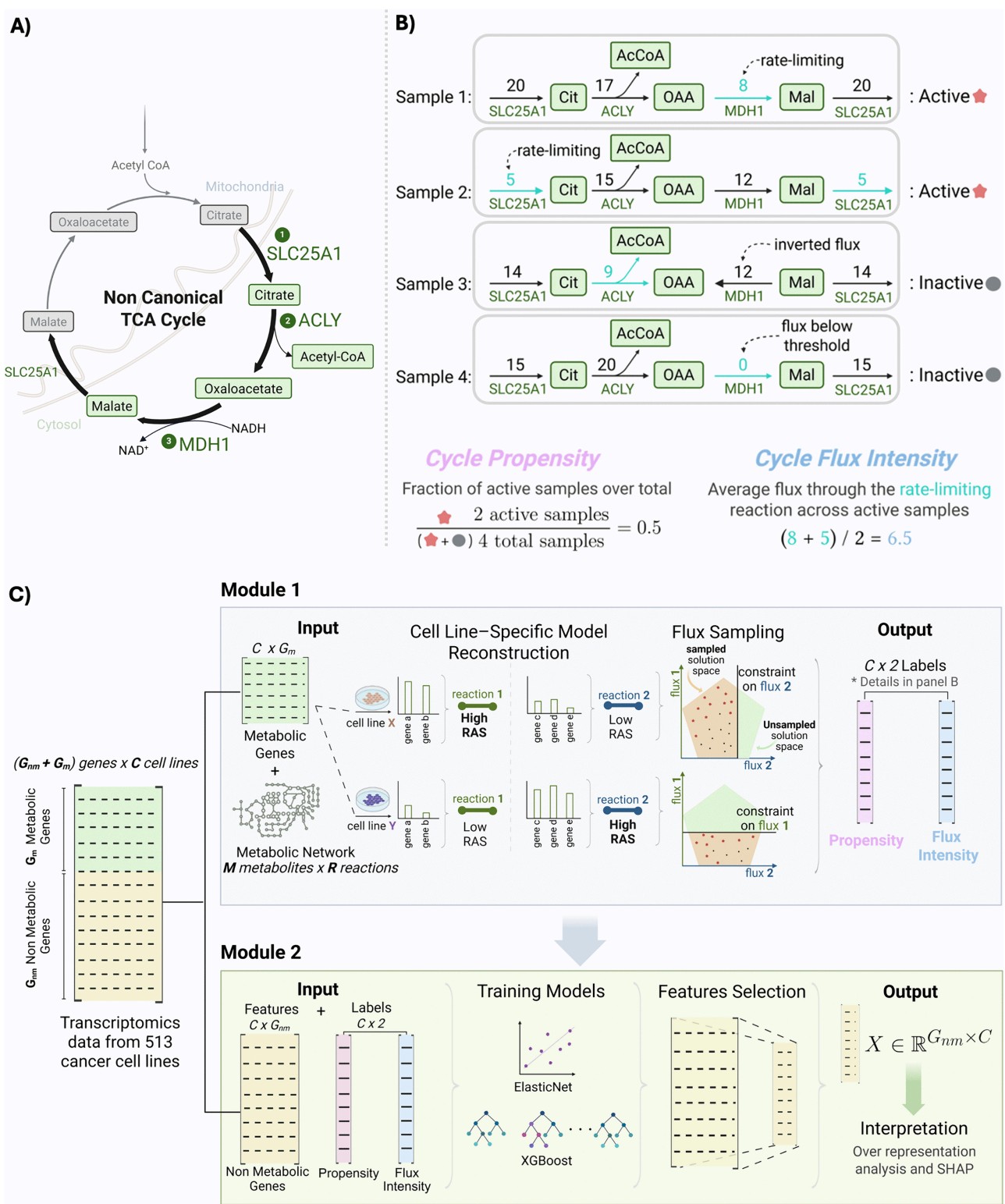

**Fig 1. Schematic representation illustrating the Cit-Mal Cycle and the computational framework. (A)** The Cit-Mal Cycle involves (1) export of mitochondrial citrate (Cit) to the cytosol via SLC25A1; (2) cleavage of citrate by ATP citrate lyase (ACLY) to produce acetyl-CoA (AcCoA) and oxaloacetate (OAA); (3) reduction of oxaloacetate to malate (Mal) and concomitant oxidation of NADH to NAD$^+$ by cytosolic malate dehydrogenase (MDH1), after which malate can be re-imported into the mitochondria via SLC25A1, thus completing the cycle. **(B)** *Cycle Propensity* is the fraction of active flux

states across all samples, whereas *Cycle Flux Intensity* is the average flux through the rate-limiting reaction among ACLY, SLC25A1, and MDH1 computed over active samples. Examples of active and inactive flux configurations are shown at the top (threshold $10^{-6}$; steady-state flux in arbitrary units, a.u.). **(C)** Module 1 integrates metabolic gene expression with a constraint-based metabolic model to infer the activity of the non-canonical TCA cycle (Cit-Mal Cycle), quantified by two complementary metrics: *Cycle Propensity* and *Cycle Flux Intensity*. These metrics are computed for each cell-specific metabolic model and used as prediction labels in Module 2, where machine learning models are trained on non-metabolic gene expression to identify transcriptional predictors of pathway activity through feature selection and SHAP-based interpretation.

Although its importance in stem cell biology is increasingly recognized, the regulation and function of the Cit-Mal Cycle in cancer remain poorly understood. In particular, it is unclear whether this cycle merely represents a metabolic adaptation or plays a broader role in promoting malignant traits. To address this knowledge gap, and to investigate the potential selective advantages that this alternative metabolic route may confer to cancer cells, we aim to systematically characterize the transcriptional programs associated with the Cit-Mal Cycle activity across diverse tumor types. However, direct measurements of compartment-specific metabolic fluxes are lacking in most cancer datasets, making it difficult to infer the activity of pathways that span multiple cellular compartments and require directional consistency, as in the Cit-Mal Cycle. In this context, constraint-based modeling (CBM) offers a particularly well-suited computational approach to estimate intracellular fluxes from gene expression profiles [2]. Nevertheless, these methods are inherently limited to metabolic genes and fail to capture regulatory or downstream transcriptional programs outside the metabolic network [3,4].

To overcome the above limitations, we developed a mechanistically informed machine learning (ML) framework that integrates CBM with supervised learning. We first applied CBM to derive quantitative metrics of the Cit-Mal Cycle activity across cancer cell lines. We then used these metrics as labels in a supervised learning problem to select transcriptional features predictive of the Cit-Mal Cycle activity. We applied our framework to over 500 cancer cell lines, revealing robust transcriptional programs and genetic vulnerabilities associated with the Cit-Mal Cycle activity. Beyond serving as labels for machine learning, our *in silico* metrics also uncovered previously unrecognized connections between the Cit-Mal Cycle activity and broader patterns of metabolic reprogramming in cancer, offering insights into how cancer cells exploit metabolic flexibility to sustain aggressive phenotypes.

## 2 Results

### 2.1 Overview of the computational framework

We conceived a two-step computational strategy integrating a metabolic modeling and a machine learning module to dissect the activity of the Cit-Mal Cycle across cancer samples.The strategy is schematized in Fig 1 and briefly summarized in the following.

The metabolic modeling module employs state-of-the-art constraint-based modeling techniques to infer the activity of the Cit-Mal Cycle directly from gene expression data [2,5]. Constraint-based models represent cellular metabolism as a network of biochemical reactions subject to stoichiometric mass balance and physicochemical constraints, such as minimal and maximal reaction fluxes [6]. These flux boundaries can be personalized using omics data—such as transcriptomic or proteomic profiles—to reflect the molecular context of individual samples [7]. In this study, we tailored the boundaries of the generic core metabolic network ENGRO2.2 based on the transcriptomes of 513 cancer cell lines cultured under standardized conditions. Specifically, we computed Reaction Activity Scores (RAS) from gene expression using Gene–Protein–Reaction (GPR) rules [8], and used these scores to modulate the upper bounds of each reaction's feasible flux, following the same strategy described in [2,9].

Each reconstructed model defines the space of feasible steady-state flux distributions for a specific cell line. While classical CBM approaches typically select a single flux vector by optimizing a biological objective (e.g., biomass yield), flux sampling has become an established alternative for systematically exploring the entire solution space consistent with

network structure and molecular constraints [10–12]. Accordingly, we employed the corner-based sampling algorithm proposed in [13] to generate thousands of feasible flux distributions per model.

To this end, we devised two complementary metrics to quantify Cit-Mal Cycle activity from the sampled flux distributions: (i) *Cycle Propensity*, defined as the fraction of sampled steady-state flux distributions in which all three hallmark reactions (SLC25A1, ACLY, MDH1) are simultaneously active. This metric is dimensionless, ranges from 0 to 1, and reflects the likelihood that the cycle operates under the given constraints; (ii) *Cycle Flux Intensity*, defined as the average flux through the bottleneck reaction in states where the cycle is active. This metric is reported in arbitrary units. Although the scale is numerically similar to conventional flux units (e.g., mmol/gDW/h), the use of relative transcriptomic constraints prevents absolute interpretation. Therefore, we explicitly treat these values as arbitrary units.

The machine learning module uses these in silico–derived labels as training targets for supervised regression models designed to predict *Cycle Propensity* and *Cycle Flux Intensity* from non-metabolic gene expression data. After model training, we applied feature selection to identify robust transcriptional predictors of Cit-Mal Cycle activity, thereby revealing broader programs potentially regulating or co-occurring with its engagement. Finally, we employed SHAP analysis to quantify the contribution of individual genes to model predictions, providing a biologically interpretable link between transcriptional features and Cit-Mal Cycle activity.

## 2.2 Flux metrics recapitulate the preferential engagement of the Cit-Mal Cycle in proliferative cells

While the transcriptionally informed flux sampling strategy we used has previously demonstrated its ability to capture cancer metabolic programs [2,5], the two flux-based metrics we devised to approximate Cit-Mal Cycle activity represent a methodological innovation. We therefore sought to assess whether these measures are consistent with experimentally derived evidence of pathway activity.

In particular, the original study by Arnold-Jackson et al. demonstrated that proliferative cells—such as cancer and embryonic stem cells—preferentially activate the Cit-Mal Cycle, whereas its engagement diminishes in differentiated and less proliferative cells [1,14,15]. We therefore tested whether these flux-derived metrics reproduce the experimentally observed increase in the Cit-Mal Cycle activity characteristic of proliferative cell states.

Since the panel of 513 cancer cell lines lacks matched normal controls, we leveraged spatial transcriptomics data from clear cell renal cell carcinoma (ccRCC) and adjacent normal tissue to assess whether our flux-based metrics capture the expected metabolic behavior of proliferative tumor regions. To this end, we computed our flux-based metrics from transcriptionally informed flux distributions previously generated in [5] from these spatial profiles, which reproduced the characteristic increase in biomass synthesis flux observed in tumor regions. We remind that flux values and derived metrics (e.g., *Cycle Flux Intensity*) are suitable for cross-sample comparisons, but not for interpreting absolute metabolic rates.

As expected, both cycle-specific metrics resulted significantly elevated in tumor areas (Fig 2), with increased *Cycle Propensity* ($t = 39.61$, $p = 7.3 \times 10^{-178}$) and *Cycle Flux Intensity* ($t = 96.30$, $p < 10^{-308}$), indicating enhanced engagement of the Cit-Mal Cycle in cancer cells.

These results demonstrate that our flux-derived metrics faithfully recapitulate the preferential engagement of the Cit-Mal Cycle in proliferative, tumor-associated regions, in agreement with its experimentally observed activation in rapidly dividing cells.

## 2.3 Tumor-specific usage and metabolic rerouting of the Cit-Mal Cycle

Having validated that our metrics capture the expected metabolic behavior in tumor-normal interface samples, we next applied them to the full pan-cancer compendium of 513 cancer cell lines to systematically characterize tumor-specific patterns of Cit-Mal Cycle activity.

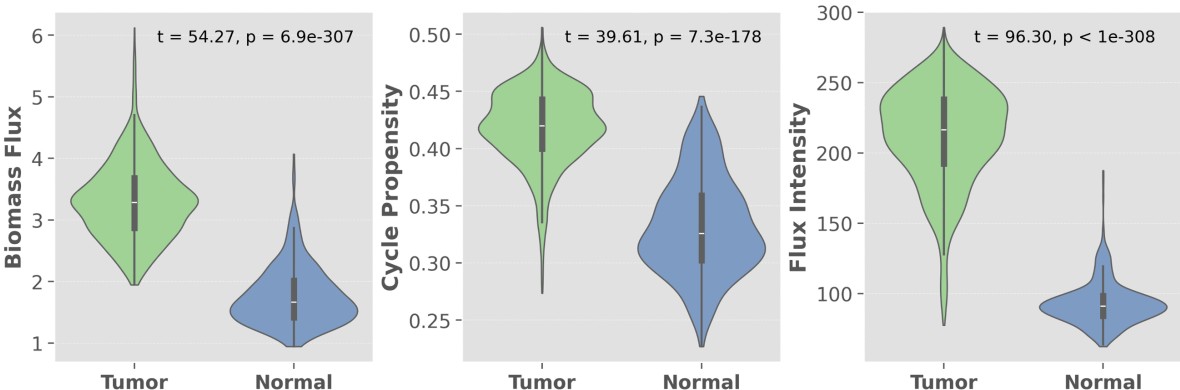

**Fig 2. Comparison of flux metrics between tumor and normal regions.** Violin plots showing the distribution of biomass pseudo-reaction flux (a.u.), *Cycle Propensity*, and *Cycle Flux Intensity* (a.u.) across tumor and adjacent normal spots in spatial transcriptomics data from ccRCC. Violins represent full data distributions with median and interquartile range. Statistical significance was assessed using a two-sided Student's *t*-test.

Despite moderate overall variance across the full cohort (0.48 $\pm$ 0.08 for *Cycle Propensity*; 12.78 $\pm$ 4.12 for *Cycle Flux Intensity*), we observed striking heterogeneity across tumor types, with distinct, tissue-specific patterns of Cit-Mal Cycle activity (Fig 3A).

Lung, stomach, breast, and pancreatic cell lines exhibited wide variability in both *Cycle Propensity* and *Cycle Flux Intensity*, indicating substantial within-type diversity. In contrast, skin, lymphoid, and esophageal tumors formed more compact clusters characterized by high *Cycle Propensity* but low *Flux Intensity*, while central nervous system and myeloid tumors showed intermediate levels of both metrics, suggesting moderate Cit-Mal Cycle activity.

Although *Cycle Propensity* and *Cycle Flux Intensity* were defined to capture complementary aspects of Cit-Mal Cycle behavior, they were found to be negatively correlated ($\rho = -0.63$, $p = 1.25 \times 10^{-58}$; Fig 3B). This inverse relationship indicates that increased likelihood of the Cit-Mal Cycle engagement tends to coincide with reduced metabolic through-put, rather than a proportional enhancement of flux. To gain mechanistic insight into this unexpected anticorrelation, we considered that *Cycle Flux Intensity* was defined based on the flux through the pathway's rate-limiting step. We therefore examined which reaction constrained flux across all cell lines. In all models, ACLY was consistently identified as the bottleneck, catalyzing the conversion of cytosolic citrate into acetyl-CoA and oxaloacetate.

Given this observation, we next investigated how citrate flux is redistributed in cell lines with higher *Cycle Propensity* (Fig 3C). As an initial step, we asked whether increased *Cycle Propensity* coincides with enhanced mitochondrial export of citrate. Indeed, we found a significant positive correlation between *Cycle Propensity* and flux through SLC25A1—the mitochondrial citrate transporter ($\rho = 0.49$, $p = 4.26 \times 10^{-32}$; Fig 3D)—suggesting that greater pathway engagement is associated with increased cytosolic citrate export. However, rather than being channeled through ACLY—the rate-limiting step—this exported citrate appears to be rerouted toward alternative cytosolic pathways. To identify these pathways, we focused on the alternative branch that uses cytosolic citrate as a substrate in the ENGRO2.2 network, namely IDH1. Indeed, the flux through IDH1 - which converts isocitrate to $\alpha$KG - showed a strong positive correlation with *Cycle Propensity* ($\rho = 0.57$, $p = 2.69 \times 10^{-46}$; Fig 3D). This flux is preceded by the activity of ACO1, which converts citrate to isocitrate, suggesting a rerouting of cytosolic citrate toward oxidative metabolism via the ACO1–IDH1 axis. To specifically capture this rerouting in the context of the Cit-Mal Cycle engagement, flux values for SLC25A1, ACO1, and IDH1 were computed exclusively from sampled states in which the cycle was active.

Together, these findings suggest that, in cell lines exhibiting a preference for the Cit-Mal Cycle engagement, cytosolic citrate may not be exclusively funneled through ACLY for acetyl-CoA production. Instead, a substantial portion may be diverted toward oxidative reactions mediated by IDH1, a well-known source of cytosolic $\alpha$KG and NADPH [16,17].

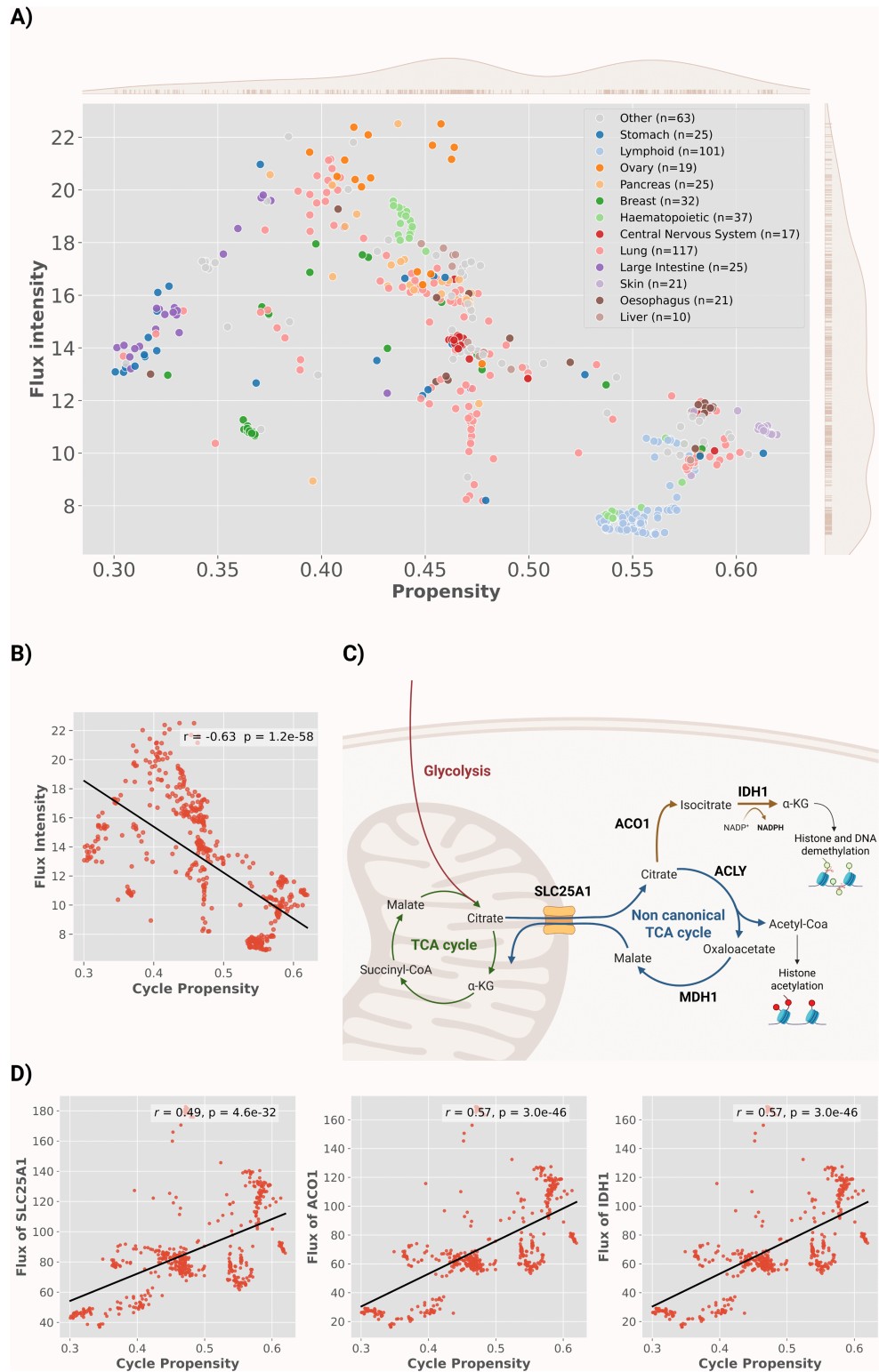

**Fig 3**. **Heterogeneous usage and routing of the non-canonical TCA cycle in cancer cell lines. (A)** Scatterplot of *Cycle Propensity* versus *Cycle Flux Intensity* (a.u.) across 513 cancer cell lines, color-coded by tissue type. Rug plots and marginal kernel density estimates (KDEs) indicate the distributions of both variables along the respective axes. Sample sizes per lineage are shown in the legend; lineages with fewer than 10 cell lines are grouped under "Other" (grey), comprising: Upper aerodigestive tract (n=9), Kidney (n=8), Autonomic ganglia (n=8), Biliary tract (n=8), Bone (n=6),

Pleura (n=5), Endometrium (n=5), Urinary tract (n=4), Prostate (n=3), Thyroid (n=3), Soft tissue (n=3), and Salivary gland (n=1). **(B)** *Cycle Propensity* and *Cycle Flux Intensity* (a.u.) are negatively correlated ($r = -0.63$, $p = 1.25 \times 10^{-58}$). **(C)** Schematic representation of cytosolic citrate routing, illustrating the bifurcation between ACLY-mediated acetyl-CoA production and the ACO1–IDH1 axis generating $\alpha$KG and NADPH. **(D)** Scatterplots showing positive correlations between *Cycle Propensity* and flux through SLC25A1 (left), ACO1 (middle), and IDH1 (right) across cancer cell lines. Flux values were computed from sampled steady states in which the Cit–Mal Cycle was active and are expressed in arbitrary units. Each point represents a single cell line and black lines indicate the fitted regression. Correlation coefficients and *p*-values were computed using Pearson's correlation.

This rerouting, combined with the persistent bottleneck at ACLY, provides a mechanistic explanation for the observed anti-correlation between *Cycle Propensity* and *Cycle Flux Intensity* and highlights how the Cit-Mal Cycle represents only one configuration within a more flexible program of metabolic shift that also includes oxidative branching via ACO1 and IDH1. Collectively, these results indicate that the Cit-Mal Cycle represents a specific manifestation of a broader non-canonical TCA configuration, which may variably engage alternative cytosolic reactions depending on cellular context. These findings also underscore the importance of jointly interpreting both metrics. While *Cycle Flux Intensity* captures the effective flux through the ACLY-limited Cit-Mal Cycle, *Cycle Propensity* reflects a broader transcriptional predisposition toward cytosolic citrate metabolism, including alternative routing through ACO1 and IDH1. Considering both metrics together thus yields a more comprehensive view of this metabolic rewiring.

## 2.4 Non-canonical TCA cycle activity associates with Warburg-like metabolism

Proliferating cells frequently adopt the Warburg effect – a metabolic reprogramming where glycolysis dominates over oxidative phosphorylation even under oxygen-rich conditions, resulting in elevated lactate secretion [18,19]. This adaptation fuels rapid biomass generation by providing both energy and biosynthetic precursors [20]. To investigate how the activity of the non-canonical TCA cycle relates to this metabolic state, we analyzed its association with key Warburg hallmarks, including oxygen consumption, lactate secretion, and the lactate-to-glucose (Lac/Glc) ratio.

For each cell line, we computed the average oxygen uptake, cytosolic lactate export, and the Lactate/Glucose ratio across all sampled steady-state flux distributions. Strikingly, higher non-canonical *Cycle Propensity* strongly correlated with reduced oxygen utilization ($\rho = -0.79$, $p = 6.2 \times 10^{-110}$) and increased lactate secretion ($\rho = 0.74$, $p = 5.5 \times 10^{-91}$). This elevation in lactate production was not merely a consequence of higher glucose uptake, as the Lac/Glc ratio also showed a strong positive correlation ($\rho = 0.71$, $p = 6.7 \times 10^{-80}$), despite unconstrained oxygen availability in simulations (Fig 4A).

As we have already emphasized above, the transcriptionally informed flux sampling strategy used to derive our non-canonical TCA cycle metrics has previously proven effective in capturing differences between breast cancer cell lines [2] and between tumor regions [5]. However, this methodology is not designed to estimate absolute flux values. As a consequence, the predicted flux for a given cell line may vary depending on the broader dataset context. For example, when the same breast cancer cell lines are analyzed within a pan-cancer dataset, their inferred fluxes may differ from those obtained in a breast-specific dataset. This is an inherent limitation of using relative transcriptomic constraints: the presence of highly dissimilar samples can compress or skew differences between more similar ones. Nevertheless, we expect that the *relative ordering* of cell lines with respect to their metabolic profiles should be preserved. To test this hypothesis and further explore the relationship between Warburg metabolism and non-canonical TCA cycle activity, we focused on a subset of three breast cancer cell lines for which the Warburg phenotype has been well characterized in the literature through experimental measurements of nutrient exchange rates: MDA-MB-231, MDA-MB-453, and MDA-MB-361. Among these, MDA-MB-231 is widely recognized as a prototypical high-Warburg cell line, whereas MDA-MB-453 and MDA-MB-361 display a weaker Warburg phenotype, with lower lactate secretion and glycolytic activity [2,21,22].

Consistent with expectations, our simulations showed that MDA-MB-231 had markedly higher non-canonical cycle activity than the other two lines (*Cycle Propensity*: 0.458 vs 0.367 and 0.365; *Cycle Flux Intensity*: 15.73 vs 10.74 and

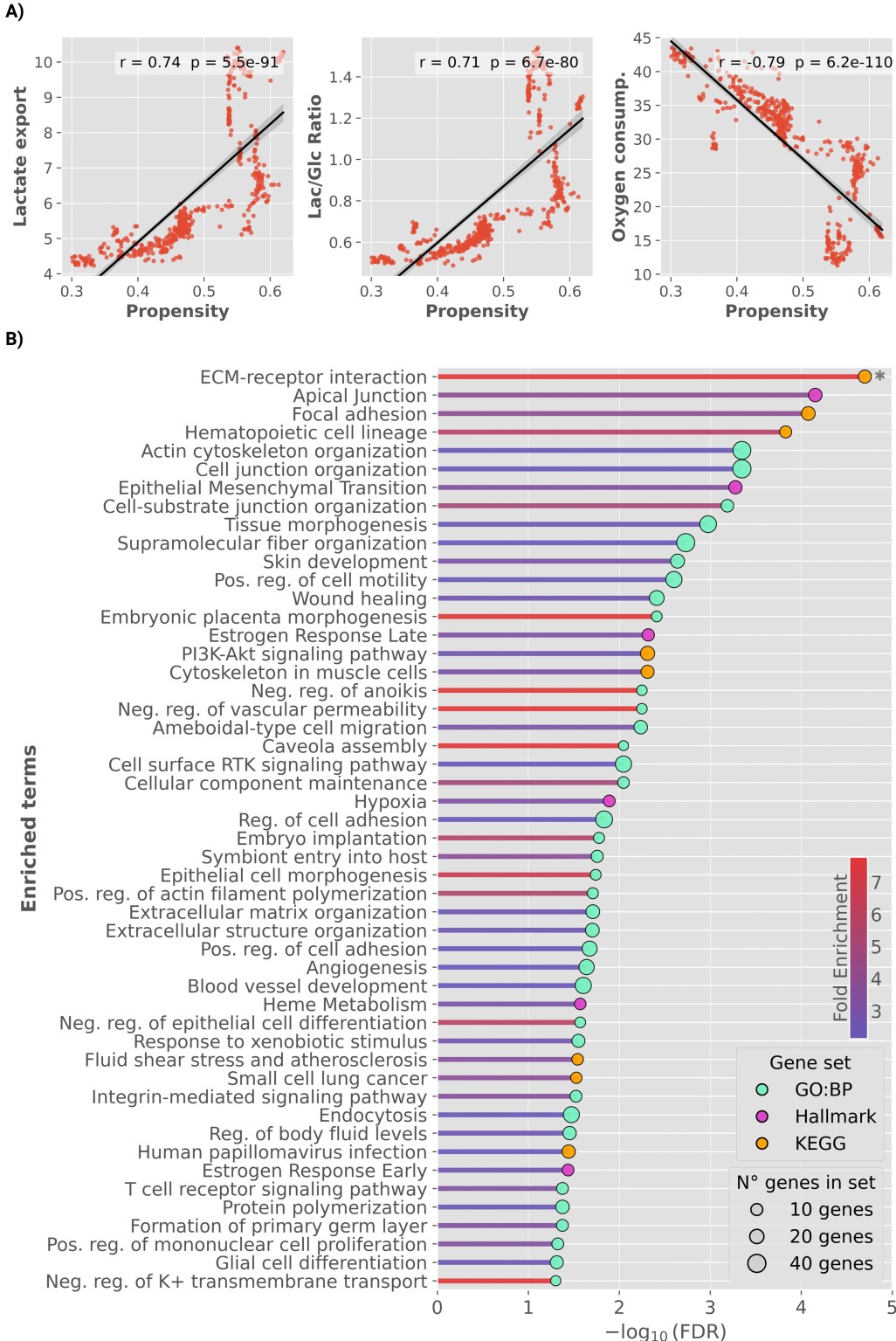

**Fig 4. Non-canonical TCA cycle activity links with Warburg metabolism and pro-malignant transcriptional programs. (A)** Relationship between *Cycle Propensity* and key metabolic markers of the Warburg effect across cancer cell lines. *Left*: *Cycle Propensity* positively correlates with lactate secretion ($\rho = 0.74$, $p = 5.5 \times 10^{-91}$); *middle*: a similar trend is observed for the lactate-to-glucose (Lac/Glc) ratio ($\rho = 0.71$, $p = 6.7 \times 10^{-80}$); *right*: conversely, a strong negative correlation is observed with oxygen consumption ($\rho = -0.79$, $p = 6.2 \times 10^{-110}$), despite unconstrained oxygen availability

in the simulations. Flux values are expressed in arbitrary units. Correlation coefficients and *p*-values were computed using Pearson's correlation. **(B)** Over-representation analysis of 436 genes predictive of *Cycle Propensity* and *Cycle Flux Intensity*, identified through ElasticNet and XGBoost models. Enriched terms are derived from GO Biological Process (GO:BP), KEGG, and MSigDB Hallmark gene sets. Bars represent enrichment significance expressed as $-\log_{10}$(FDR); circles are scaled by the number of genes in each set and color-coded by gene set source, while bar colors reflect fold enrichment. The asterisk (*) denotes the term *ECM–receptor interaction* (FDR $= 1.13 \times 10^{-7}$) truncated for visualization purposes.

10.93 a.u.). Regarding Warburg-related fluxes, MDA-MB-231 also exhibited slightly higher lactate secretion (5.61 vs 5.17 and 5.13 a.u.) and lactate-to-glucose ratios (0.69 vs 0.67 and 0.65), while oxygen uptake was similar across the three lines (30.75 vs 29.26 and 30.86 a.u.).

These modest differences likely reflect the intermediate positioning of breast cancer cells within the broader metabolic landscape captured by our pan-cancer dataset. Nonetheless, within this constrained dynamic range, our simulations reproduced the expected hierarchy of Warburg activity, with MDA-MB-231 exhibiting the most glycolysis-dominant metabolic profile.

The robust association between higher *Cycle Propensity* and enhanced Warburg traits supports the view that engagement of the non-canonical TCA cycle accompanies a metabolic shift toward glycolytic metabolism and reduced oxidative activity. Consistent with recent surveys and with the original hypothesis of Arnold-Jackson *et al.* [1,23], activation of cytosolic citrate metabolism enables cells to regenerate cytosolic NAD⁺ to sustain glycolysis, while limiting mitochondrial NADH production and thus lowering the requirement for oxygen-dependent NADH reoxidation.

## 2.5 Non-canonical TCA cycle activity defines a malignancy-linked transcriptional landscape

To uncover non-metabolic transcriptional programs associated with non-canonical TCA cycle activity, we trained multi-output regression models that simultaneously predict *Cycle Propensity* and *Cycle Flux Intensity* using a single model fitted on non-metabolic gene expression profiles. Because these two metrics are complementary yet correlated, a joint multi-output formulation enables the model to leverage shared predictive structure and identify common transcriptional drivers. Specifically, we employed a multitask ElasticNet model, which captures shared linear associations between gene expression and target variables, and a multi-output XGBoost, a tree-based approach capable of capturing more complex nonlinear relationships.

Both models showed good predictive performance. The multitask ElasticNet reached an average $R^2$ of $0.778 \pm 0.047$ for *Cycle Propensity* and $0.768 \pm 0.038$ for *Cycle Flux Intensity*, whereas the multi-output XGBoost achieved $0.813 \pm 0.047$ and $0.806 \pm 0.024$, respectively. Detailed prediction performance across folds is reported in S1 and S2 Figs. Although our primary aim is to use these models to identify transcriptional programs associated with non-canonical TCA cycle activity rather than to build universal predictors, we nonetheless adopted learning strategies specifically designed to limit overfitting and assessed their behaviour through learning curves (S3 and S4 Figs).

For each model, we selected genes consistently identified as predictive across outer folds. Applying this criterion, we identified 322 stable features from the MultiTask ElasticNet model, 186 from the multi-output XGBoost model, and 72 genes consistently selected by both methods. The complete list of 436 selected features is provided in S1 Table.

We next assessed whether the 436 genes consistently predictive of non-canonical TCA activity form functionally coordinated programs by constructing a protein–protein interaction (PPI) network. Of the 436 genes, 361 are mapped to the PPI network, forming 229 edges—significantly more than the 78 edges expected by random chance ($p < 10^{-16}$); the full network map is provided in S5 Fig. Clustering of the PPI network identified three main modules (S6 Fig). The largest cluster (21 genes) was enriched for *extracellular matrix (ECM) receptor interaction*, the second cluster (18 genes) for the *RHO GTPase cycle*, and the third cluster (10 genes) for *RUNX3–YAP1–mediated transcription* and *transcriptional regulation of granulopoiesis*.

To systematically characterize the biological functions represented within the 436-gene signature, we performed Over-Representation Analysis (ORA) across GO Biological Process, KEGG, and MSigDB Hallmark gene sets, using 18,082 genes as background. Enrichment significance was evaluated with Benjamini–Hochberg correction, and only terms with false discovery rate (*FDR*)<0.05 were considered significant (Fig 4B). Detailed enrichment statistics for all significant terms, are reported in S2 Table.

The selected genes showed consistent enrichment for processes associated with invasive and metastatic behavior. Enriched terms included *epithelial–mesenchymal transition* (EMT), *Apical junction*, *cell–substrate junction organization*, and *positive regulation of cell motility*. In addition, enrichment for *negative regulation of anoikis*—a form of apoptosis triggered by the loss of cell–matrix attachment—suggests enhanced resistance to detachment-induced cell death [24]. Collectively, these findings indicate that activation of the non-canonical TCA cycle is associated with transcriptional programs enabling an invasive and metastatic tumor phenotype.

Beyond invasion, the enriched genes also reflected transcriptional programs related to embryonic and dedifferentiation-associated programs, including *formation of primary germ layer*, *embryonic placenta morphogenesis*, *embryo implantation*, *tissue morphogenesis*, and *negative regulation of epithelial cell differentiation*. These enrichments collectively point to the reactivation of developmental programs normally restricted to embryogenesis, suggesting that cells with higher non-canonical TCA cycle activity may acquire a more plastic, stem-like transcriptional state [25]. This observation is consistent with experimental evidence showing that non-canonical TCA metabolism accompanies transitions in cell fate [1].

We also observed enrichment for pathways related to hypoxia and angiogenesis, two processes tightly interconnected, as hypoxia is a major inducer of pro-angiogenic signaling [26].

At the signaling level, we observed enrichment for several oncogenic pathways. Notably, *PI3K–Akt signaling* and *cell surface receptor protein tyrosine kinase signaling*, were significantly enriched, reflecting the engagement of growth factor–driven pathways that sustain proliferation and survival. Mechanistically, receptor tyrosine kinases converge on the PI3K–Akt axis, promoting cell-cycle progression and anabolic metabolism—pathways frequently hyperactivated across cancers [27]. Furthermore, hormone-responsive transcriptional modules, including *Estrogen Response Late* and *Estrogen Response Early*, suggest involvement of receptor-mediated programs linked to hormonal signaling.

Collectively, these results reveal a robust transcriptional landscape associated with non-canonical TCA cycle activity, encompassing key hallmarks of malignancy—including metastatic behavior, angiogenesis, stemness, and key oncogenic signaling.

## 2.6 SHAP highlights oncogenic and tumor suppressor genes as top predictors of non-canonical TCA activity

To interpret the two cycle-activity metrics at single-gene resolution for the stably selected genes, we use SHAP values to derive local, cell-line–specific attributions, thereby identifying the most informative genes and quantifying their contributions in each line. This approach provides not only a *global* ranking of predictive genes, but also a map of which genes are decisive *in which* cell lines, thereby guiding the prioritization of targeted wet-lab hypotheses (joint gene × cell-line selection).

To identify a robust setting for SHAP analysis, we first compared the predictive performance of regression models trained with different panels of selected features. In particular, we evaluated the union of genes stably selected across cross-validation folds by at least one model (ElasticNet or XGBoost), as well as the model-specific panels selected exclusively by each algorithm.

The union panel yielded the best overall performance, achieving high accuracy across both regression targets (*Cycle Propensity* and *Cycle Flux Intensity*) with both models. ElasticNet achieved $\overline{R^2}_{\text{Propensity}} = 0.858 \pm 0.033$, $\overline{R^2}_{\text{Intensity}} = 0.864 \pm 0.027$; while XGBoost obtained $\overline{R^2}_{\text{Propensity}} = 0.862 \pm 0.025$, $\overline{R^2}_{\text{Intensity}} = 0.868 \pm 0.030$. Importantly, these performance metrics were used solely to compare feature panels, and should not be interpreted as estimates of generalization

ability. Because feature selection was performed on the full dataset across outer folds, the resulting scores may be over-optimistic.

Notably, the *union* panel was the top-performing configuration for ElasticNet and one of the best for XGBoost. For the latter, predictive performance on the *union* panel was statistically equivalent to that obtained using its full feature set *(Cycle Propensity*: FDR = 0.144; Cycle Intensity: FDR = 1.0). These results indicate that the *union* panel preserves predictive accuracy while offering improved interpretability. We therefore selected it for downstream SHAP analysis to enable a fair comparison between models and to derive biologically meaningful insights.

Because both models performed comparably yet selected partly distinct feature sets, we retained both ElasticNet and XGBoost for downstream SHAP interpretation to capture complementary aspects of gene-level importance. Top-20 genes per target and model are shown in Fig 5, while the stability of gene-level importance is summarized in S7 and S8 Figs. Complete per–cell-line SHAP attributions are provided in S1 Data.

Two genes recur among the top contributors across both targets and both models—pleckstrin homology domain-containing family A member 6 (*PLEKHA6*) and Selenium-Binding Protein 1 (*SELENBP1*). *SELENBP1* is reported as a tumor suppressor frequently downregulated across multiple cancers and has been linked to redox homeostasis and lipid/glucose metabolism in vivo [28,29]. *PLEKHA6* encodes a PH-domain, $PIP_2$/$PIP_3$-binding protein. In LUAD, *PLEKHA6* silencing curtails proliferation, slows migration, and reduces clonogenic growth, accompanied by a decrease in $\beta$-catenin levels—consistent with upstream control of WNT/$\beta$-catenin signaling [30]. Concordantly, *FZD5*—a WNT receptor able to transduce canonical ($\beta$-catenin) or non-canonical ($Ca^{2+}$) signals depending on ligand and cellular state—also emerges among the XGBoost-selected top-predictors for both targets [31]

For *Cycle Propensity*, both models jointly select *SMPDL3B*, *ID4*, and *HID1*. *SMPDL3B* (sphingomyelin phosphodi-esterase acid-like 3B) supports sphingolipid remodeling and has functional evidence for roles in motility/proliferation in prostate, acute myeloid leukemia, and ovarian cancer [32–34]. *ID4* (inhibitor of DNA binding 4) is a DNA-binding–deficient bHLH regulator modulating lineage specification and developmental programs. In cancer, it is most often tumor-suppressive via epigenetic silencing, yet shows context-dependent oncogenic activity in select subtypes [35]. *HID1* (HID-1 domain–containing protein) is a conserved trans-Golgi/secretory-granule factor required for dense-core vesicle maturation and regulated secretion; direct oncologic evidence is limited [36]. For *Cycle Flux Intensity*, *EPHB4* and *ERFE* emerge as shared predictors across models; *EPHB4* (EPH receptor B4) orchestrates vascular development and tissue remodelling; in oncologic contexts, it frequently connects to angiogenesis and microenvironment-driven tumor progression [37,38]. *ERFE* (erythroferrone) is a secreted erythroid hormone that tunes iron homeostasis, in silico pan-cancer analyses indicate broad overexpression linked to poorer outcomes [39].

Taken together, the SHAP-prioritized features are functionally implicated in diverse aspects of tumorigenesis, including metabolic reprogramming, proliferative control, angiogenic and microenvironmental remodeling and cell motility. These convergent signals provide a robust basis for targeted experimental validation and hypothesis-driven perturbations in future work.

## 2.7 Genetic dependencies associated with non-canonical TCA cycle activity

We finally investigated whether variations in non-canonical TCA cycle activity are associated with specific genetic vulnerabilities. To this end, we assessed the correlation between our cycle activity metrics and DepMap CERES gene–effect scores, where lower values indicate stronger gene essentiality [40]. Correlations were computed across all assayed genes, using residualized values for both activity metrics and gene dependency profiles to control for confounding effects of proliferation, *TP53* status, and histological subtype.

We identified 28 genes whose dependency profiles significantly correlated with *Cycle Propensity*, and 3 genes significantly associated with *Cycle Flux Intensity*.

Regarding *Cycle Propensity* among all significant genes (Figs 6A and S9), *SLC2A1*, which encodes the Glucose transporter 1, showed the strongest correlation with ($\rho = -0.29$, FDR $= 3.1 \times 10^{-3}$). GLUT1 mediates basal glucose uptake and

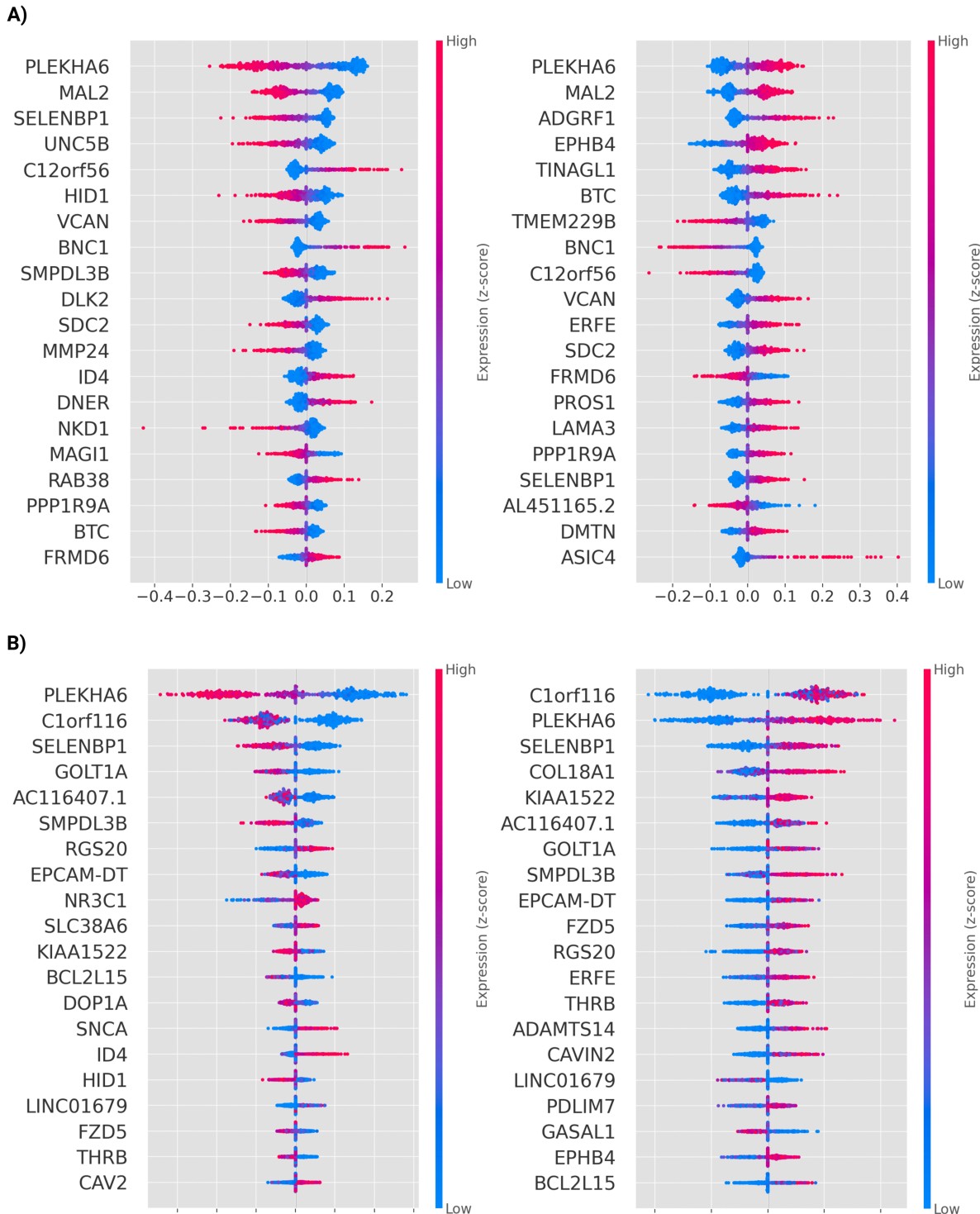

**Fig 5**. **SHAP feature importance across models and cycle-activity metrics.** Each panel shows SHAP beeswarm summary plots, where the x-axis represents SHAP values (feature contribution to prediction) and the color scale indicates normalized gene expression (Z-score). Each dot corresponds to a single cancer cell line, and genes on the y-axis are ranked by their overall contribution to the model output (mean absolute SHAP value). **(A)** Results for the Multi-Task ElasticNet model, showing the top 20 genes predictive of *Cycle Propensity* (left) and *Cycle Flux Intensity* (right). **(B)** Results for the Multi-Output XGBoost model, showing the top 20 genes predictive of *Cycle Propensity* (left) and *Cycle Flux Intensity* (right).

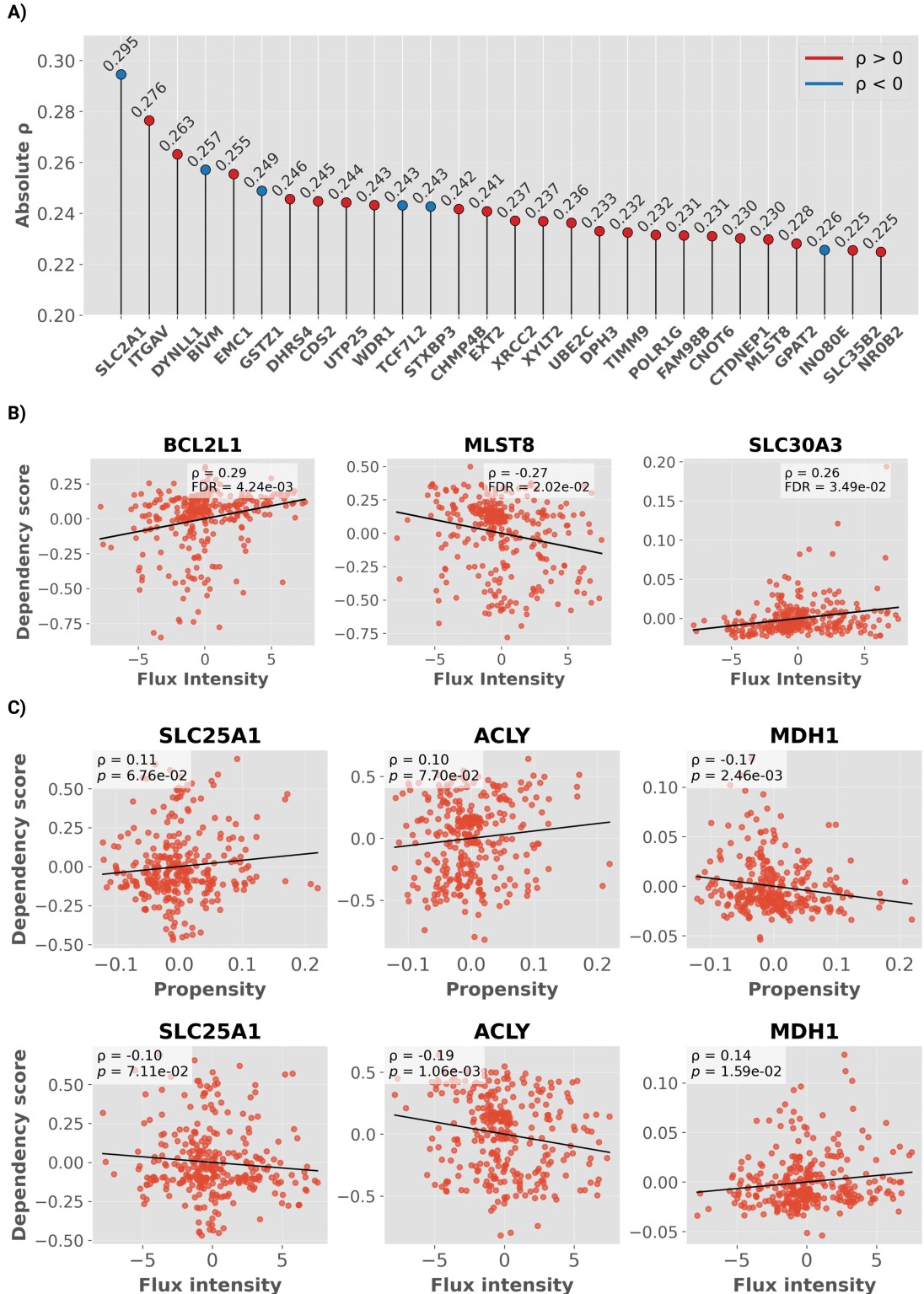

**Fig 6. Gene-dependency landscape associated with non-canonical TCA cycle metrics. (A)** Lollipop plot showing the absolute correlation coefficients between residualized gene–dependency scores and residualized *Cycle Propensity* across all significant genes. Red and blue colors denote positive and negative correlations, respectively. Correlation coefficients and *p*-values were computed using Spearman's correlation on residualized values. **(B)** Correlation between residualized *Flux Intensity* (a.u.) and residualized gene–dependency scores for significantly associated genes identified

in the genome-wide analysis (FDR <0.05). Each point represents a cancer cell line; black lines indicate the fitted regression. **(C)** Correlation between residualized *Cycle Propensity* and residualized *Flux Intensity* (a.u.) with residualized DepMap gene–dependency scores for enzymes catalyzing the non-canonical TCA cycle.

is widely recognized as a major facilitator of the Warburg effect, being upregulated across diverse cancer types to sustain elevated glycolytic flux and biosynthetic demand [41,42]. This negative association aligns with the Warburg-like metabolic phenotype observed earlier, suggesting that cell lines with higher engagement of the non-canonical TCA cycle exhibit enhanced glucose dependence and glycolytic activity. The remaining list of signficant correlations includes genes involved in lipid metabolism (*CDS2*, *CTDNEP1*, *GPAT2*, *DHRS4*), redox homeostasis (*GSTZ1*), and heparan sulfate biosynthesis (*EXT2*, *XYLT2*, *SLC35B2*), as well as membrane transporters such as *SLC30A3*. Other associations were found with genes mediating cell adhesion and motility (*ITGAV*, *WDR1*); protein trafficking and sorting (*STXBP3*, *CHMP4B*, *EMC1*, *DYNLL1*, *TIMM9*); RNA metabolism (*POLR1G*, *UTP25*, *CNOT6*, *FAM98B*, *DPH3*); and DNA repair (*XRCC2*, *INO80E*). Notably, the transcription factor *TCF7L2* (Transcription factor 7-like 2)—a key downstream effector of WNT/$\beta$-catenin signaling—also exhibited a modest negative association with *Cycle Propensity* ($\rho = -0.24$, FDR = 0.029), suggesting potential interplay between WNT/$\beta$-catenin transcriptional programs and non-canonical cycle activity.

Regarding *Cycle Flux Intensity*, the three identified genes were *BCL2L1*, *MLST8*, and *SLC30A1* (Fig 6B). *BCL2L1* (BCL-2–like 1) encodes the anti-apoptotic protein, a mitochondrial outer-membrane factor that inhibits caspase activation and promotes cell survival ($\rho = 0.29$, $FDR = 4.24 \times 10^{-3}$) [43]. *MLST8* (MTOR-associated protein, LST8 homolog) is a core component of both mTORC1 and mTORC2 complexes, mediating growth and metabolic signaling downstream of the PI3K–Akt pathway ($\rho = -0.27$, $FDR = 2.02 \times 10^{-2}$), which was enriched in the previous ORA [44]. *SLC30A1* (Zinc transporter 1) encodes a zinc efflux transporter that also facilitates copper uptake ($\rho = 0.26$, $FDR = 3.49 \times 10^{-2}$) [45].

Focusing on the three genes encoding the enzymes that directly catalyze the Cit-Mal Cycle (Fig 6C), neither *SLC25A1* nor *ACLY* showed a significant association with *Cycle Propensity*; in contrast, *MDH1* showed a modest inverse relationship ($\rho = -0.17$, $p = 2.46 \times 10^{-3}$). For *Cycle Flux Intensity*, *ACLY* correlated negatively ($\rho = -0.19$, $p = 1.06 \times 10^{-3}$), whereas *MDH1* displayed a weak positive association ($\rho = 0.14$, $p = 1.59 \times 10^{-2}$).

In summary, activity of the non-canonical TCA cycle is accompanied by distinct patterns of genetic dependency, outlining a specific vulnerability landscape. These findings provide a conceptual basis for therapeutic strategies aimed at targeting context-specific vulnerabilities.

## 3 Discussion

We introduced a novel integrative framework that combines constraint-based metabolic modeling with machine learning to indirectly link metabolic flux states with gene expression programs across a wide compendium of cancer cell lines. Rather than modeling gene regulation explicitly, our approach leverages flux distributions under transcriptome-derived constraints to infer functionally informative metrics of metabolic activity. We then use these metrics as continuous measures of metabolic phenotype to identify gene expression programs associated with specific metabolic configurations. In doing so, we bridge two complementary dimensions of cancer cell biology: tumor metabolic reprogramming and transcriptional programs underpinning malignant progression. Within this framework, we applied our approach to the Cit-Mal Cycle, a recently described non-canonical variant of the TCA cycle, to identify the transcriptional programs associated with its activity. While our framework does not resolve causality or directionality (i.e., whether the associated genes act as regulators or downstream effectors), it offers a powerful strategy to derive flux-informed, testable associations between metabolic rewiring and malignant traits while relying solely on transcriptomic data.

At a purely metabolic level, our study revealed that the activity of the non-canonical TCA cycle is highly heterogeneous across tumors. As the likelihood of pathway engagement (*Cycle Propensity*) increases, the effective flux through the cycle

decreases. According to our modeling results, the observed reduction in effective cycle flux is accompanied by a preferential rerouting of cytosolic citrate toward $\alpha$KG production via ACO1 and IDH1, rather than its conversion to acetyl-CoA by ACLY. This shift indicates that the non-canonical TCA cycle is more articulated than previously described by Arnold-Jackson et al., incorporating not only the canonical ACLY-driven citrate-to-acetyl-CoA route, but also at least one alternative branch that channels citrate toward $\alpha$KG production.

Beyond citrate rerouting, we observed that increased engagement of the non-canonical TCA cycle coincides with enhanced lactate secretion and reduced oxygen consumption, reproducing a Warburg-like metabolic phenotype. In particular, earlier research has demonstrated that ACLY promotes glycolytic metabolism and lactate production in cancer cells. For example, its overexpression in breast cancer enhances glycolytic enzyme expression, glucose uptake, and lactate secretion, facilitating tumor progression via EMT [46]. Similarly, in glioblastoma, ACLY accumulation in pseudopodia sustains migration and preserves glycolytic flux under mitochondrial inhibition [47]. Taken together, citrate rerouting and the associated Warburg-like metabolic phenotype suggest that the non-canonical TCA cycle is not merely an auxiliary pathway within central carbon metabolism, but rather part of a broader reorganization of the metabolic network. This reprogramming of the overall metabolic network likely reflects selective pressures to optimize carbon and redox economy in response to specific cellular and environmental contexts [1,14,15]. By retaining reduced carbon instead of oxidizing it to $CO_2$, the cycle may preserve vital intermediates for lipid, nucleotide, and amino acid synthesis. Similarly, regeneration of cytosolic $NAD^+$ and a reduced mitochondrial NADH output collectively lessen the need for oxygen-dependent NADH reoxidation, thus enabling anabolic growth under low-oxygen conditions.

At the transcriptional level, our machine-learning module identifies a set of genes associated with non-canonical TCA cycle activity that is enriched for programs linked to embryonic development and dedifferentiation. This pattern is consistent with experimental evidence from Arnold-Jackson et al. [1], who implicated the Cit-Mal Cycle in cell state transitions during developmental progression. Such transitions are tightly coupled to chromatin plasticity and epigenetic regulation: dynamic histone and DNA modifications, together with changes in higher-order chromatin organization, are now recognized as central determinants of cell fate and identity during development and reprogramming [48]. Similar principles apply to tumors, where epigenetic plasticity and chromatin disruption can reconfigure cell identity, enabling aberrant fate transitions that contribute to multiple hallmarks of cancer [49]. In line with this view, previous studies have shown that ACLY and the mitochondrial citrate transporter SLC25A1 contribute to the maintenance of cancer stem cell–like properties: knockdown of ACLY suppresses self-renewal capacity and tumorsphere formation in models of NSCLC, HCC, and breast cancer [50,51], whereas SLC25A1 overexpression in NSCLC and liver cancer stem cells promotes self-renewal, enhances the expression of canonical stemness markers, and is required for proper mitochondrial function [52,53].

Building on this, and although our model does not explicitly capture epigenetic regulation, we hypothesize that the rerouting of citrate we observe may confer not only metabolic and biosynthetic advantages, but also an epigenetic one. On one side, ACLY-derived acetyl-CoA supports histone and protein acetylation. On the other hand, the citrate-to-$\alpha$KG branch supplies $\alpha$KG, a critical cofactor for $\alpha$KG-dependent dioxygenases involved in histone and DNA demethylation. Coordinated control of flux through these two branches could therefore jointly tune acetylation and methylation dynamics on chromatin, shaping gene-expression programs associated with malignant plasticity in cancer. Previous studies have shown that cytosolic IDH1-derived $\alpha$KG modulates histone methylation patterns to control lineage commitment, such as in brown adipocyte differentiation, where $\alpha$KG suppresses adipogenesis by reducing H3K4me3 at key developmental gene promoters [54]. Moreover, mutations in *IDH1* and *IDH2*, recurrently observed in gliomas and acute myeloid leukemias, lead to the neomorphic production of 2-hydroxyglutarate (2HG), a competitive inhibitor of $\alpha$KG-dependent dioxygenases. This aberrant metabolite impairs histone demethylation, resulting in persistent repression of differentiation-associated genes and maintenance of a progenitor-like transcriptional state [55].

Beyond their association with developmental programs, the transcriptional signatures linked to non-canonical TCA cycle activity are also enriched for other hallmarks of malignancy, including metastatic behavior, angiogenesis and key

oncogenic signaling pathways. The enrichment in metastasis-related terms is consistent with experimental evidence implicating ACLY in invasion and metastatic spread. ACLY has been shown to promote colon cancer metastasis by enhancing EMT [56]. A similar pro-invasive role for ACLY has been reported in hepatocellular carcinoma (HCC), in line with its function in sustaining metastatic traits [51].

Consistently, we also observed enrichment of genes involved in oncogenic signaling, including components of the PI3K/Akt pathway. ACLY is a known downstream target of PI3K/Akt and is activated through direct Akt-mediated phosphorylation [57]. Although WNT/$\beta$-catenin signaling was not enriched at the pathway level, several components of this axis (e.g., *TCF7L2*, *FZD5*) emerged across our SHAP and gene dependency analyses, consistent with previous evidence linking ACLY to $\beta$-catenin stabilization and facilitating its translocation to the nucleus, thereby enhancing $\beta$-catenin–dependent transcription [56].

Interestingly, we also observed enrichment for angiogenesis-related programs. This pattern may reflect a coordinated metabolic and transcriptional configuration favoring hypoxia tolerance. At the metabolic level, flux analysis indicated reduced oxygen requirements in cell lines with higher cycle propensity. In parallel, the enrichment of angiogenesis-related pathways represents a transcriptional program facilitating vascular remodeling to restore oxygen delivery [58]. Together, these observations suggest coordinated metabolic and transcriptional adaptations through which non-canonical TCA activity may support cellular resilience under oxygen-limiting conditions. To the best of our knowledge, no direct mechanistic link has yet been experimentally established between enzymes of the non-canonical TCA cycle and angiogenic signaling in cancer, highlighting a potentially unexplored connection between metabolic adaptation and vascular remodeling that warrants further experimental investigation.

The hypotheses emerging from our computational analysis point to several directions for experimental validation. The predicted rerouting of cytosolic citrate toward the ACO1–IDH1 branch represents an intriguing metabolic adaptation that could be directly assessed through isotope tracing experiments, testing whether citrate-derived carbon is preferentially channeled to $\alpha$KG rather than converted to acetyl-CoA via ACLY. Likewise, the association between non-canonical TCA cycle engagement and a Warburg-like phenotype could be verified by measuring oxygen consumption rate, extracellular acidification rate, and lactate secretion across representative high-propensity/low-intensity and low-propensity/high-intensity cell lines. In this regard, our comparison of simulated flux profiles for breast cancer cell lines with experimentally characterised Warburg phenotypes already provides a preliminary consistency check, showing qualitative agreement between model-predicted Warburg-related flux and experimental observation. Although our computational framework identifies robust associations between malignancy-associated transcriptional programs and non-canonical TCA activity, it does not resolve causal regulators from downstream effectors. One potential approach to disentangle these relationships would be to measure histone acetylation at promoters of selected genes, since ACLY-derived acetyl-CoA directly contributes to chromatin acetylation; however, such epigenetic data are currently unavailable in the CCLE resource. Nonetheless, the genes identified through our machine learning–based feature selection, together with their SHAP-derived cell line–specific contributions, provide a rational foundation for selecting gene–cell line pairs for targeted experimental validation. Functional perturbation studies—such as gene knockdown or knockout—could help clarify the causal influence of these genes on non-canonical TCA cycle activity and the broader metabolic–transcriptional coupling inferred from our analysis.

Importantly, while our analysis focuses on the non-canonical TCA cycle, the modular framework developed here is broadly applicable to other metabolic pathways, or even to different organisms. However, it is essential to emphasize that the performance of our pipeline is highly dependent on the accuracy and completeness of the underlying metabolic model reconstruction. We used a well-curated metabolic core model of central carbon metabolism that has proven effective in studying cancer metabolic rewiring. When expanding our framework to other regions of the human metabolic network or other organisms, refinement of these models will be essential to ensure that the derived outputs reflect biologically meaningful and context-specific insights.

## 4 Methods

### 4.1 Data collection

The gene expression data were obtained from the Cancer Cell Line Encyclopedia (CCLE) [59], including RNA-seq profiles of 1,019 human cancer cell lines, normalized to Transcripts Per Million (TPM). To minimize experimental variability arising from heterogeneous culture conditions, we restricted the dataset to 513 cell lines cultured exclusively in RPMI-1640 medium—the most prevalent condition in the dataset.

For the validation of our cycle activity metrics, we used the fluxes generated in [5], derived from spatial transcriptomics data from ccRCC tissue and adjacent normal parenchyma (sample ID: PD45816_I2), generated with the 10x Genomics Visium platform [60]. In total, the dataset included 1,705 spatial spots, comprising 1,245 tumor spots and 460 renal parenchyma spots.

Gene dependency scores were retrieved from the Cancer Dependency Map (DepMap, *ver. 25Q3*) [40], and somatic mutation calls were obtained from the CCLE mutation dataset [61]

### 4.2 Metabolic Model

Cell line–specific models were built based on the recently published ENGRO2 core network model, a curated reconstruction of human central carbon metabolism [2]. We chose a network of reduced complexity over genome-scale reconstructions because of its greater computational efficiency and interpretability. While genome-scale models provide broader metabolic coverage, their inclusion of numerous alternative pathways can obscure specific metabolic activities in flux simulations. In line with this, preliminary flux sampling experiments using the Recon3D [62] model revealed that both canonical and non-canonical TCA cycles were poorly represented (S10 Fig). These results support the use of a core model for our analysis.

Compared to the original ENGRO2 network [2], we introduced two improvements and one essential modification. As improvements, we added reactions for galactose and cholesterol uptake, thereby expanding the range of possible nutrients considered in the simulations, and we refined the GPR rules related to amino acid uptake [63]. In the original ENGRO2 model, some uptake reactions were linked to a limited set of transporters, which caused unrealistic growth impairments in certain cell lines. To address this, we extended the GPR associations by including all known amino acid transporters identified in the Recon3D network [62], using OR logical operators to merge them. These modifications increased the biological coverage and precision of the model without directly affecting the present analysis.

In addition to these improvements, one essential modification was required for this study, namely the extension of cytosolic acetyl-CoA production pathways. In the original ENGRO2 model, ACLY was the sole source of cytosolic acetyl-CoA. Because this metabolite is required for fatty acid synthesis and thus biomass production, ACLY became an artificially essential reaction, predicted to be active in all sampled flux distributions. However, experimental evidence shows that ACLY is not strictly essential, as its loss can be compensated [64]. To better reflect biological reality, we therefore incorporated literature-supported alternative pathways for cytosolic acetyl-CoA production, which include acetyl-CoA generation from acetate via acyl-CoA synthetase short chain family member 2 (ACSS2) [64], endogenous acetate generation [65], and the acetylcarnitine shuttle, where mitochondrial acetyl-CoA is transferred to the cytosol via carnitine acetyltransferase (CrAT) [66].

This extended version, hereafter referred to as ENGRO2.2, comprises 395 metabolites, 469 reactions, and 498 genes. Among the 469 reactions, 351 are associated with GPR rules, enabling robust integration of transcriptomic data into the model. The biomass pseudo-reaction of ENGRO2 corresponds with the biomass reaction of the Recon3D model, in terms of the set of metabolites considered and their corresponding stoichiometric coefficients.

### 4.3 Cell line–specific model reconstruction

**4.3.1 Reaction activity score calculation.** Gene expression values from the CCLE dataset were log-transformed using the `log1p` function and imputed using MAGIC with default parameters. Although denoising is more commonly applied to single-cell RNA-sequencing data [63]—where sparsity arises from dropout events—we found that applying it to our bulk transcriptomic profiles substantially improved downstream analyses. In particular, it yielded higher silhouette scores when clustering cell lines based on simulated flux distributions, and also led to significantly better alignment with transcriptome-based clusters defined by metabolic gene expression. In the absence of denoising, the two clustering solutions showed low concordance; a detailed description of the MAGIC preprocessing and its benchmarking on bulk RNA-seq data is available at [67].

The resulting denoised expression matrix was then used to compute Reaction Activity Scores (RAS) for all metabolic reactions across cell lines according to their GPR rules, as described previously [8]. Logical expressions were resolved such that the minimum transcript level was used for AND logic, while the sum was used for OR logic. Standard operator precedence (AND before OR) was applied when both operators were present. The resulting RAS matrix provided the input for the subsequent reconstruction of cell line–specific metabolic models.

**4.3.2 Flux variability analysis.** To characterize the feasible flux space of the metabolic network, defined by mass balance constraints and medium composition, we employed Flux Variability Analysis (FVA) [68–70]. This constraint-based approach computes the minimum and maximum possible flux ($v_j^{min}$, $v_j^{max}$) for each reaction $j$ ($j = 1, \dots, R$) subject to the defined constraints.

For each reaction $j$, FVA solves the optimization problem:

$$\max / \min \quad v_j$$
$$\text{s.t.} \quad S\vec{v} = \vec{0} \tag{1}$$
$$\vec{v}_L \leq \vec{v} \leq \vec{v}_U$$

where $S$ is the stoichiometric matrix, $\vec{v}$ the flux vector, $S\vec{v} = \vec{0}$ enforces steady-state mass balance, and $\vec{v}_L$, $\vec{v}_U$ represent flux bounds imposed by biological constraints and the RPMI-1640 medium.

**4.3.3 Cell line–specific flux constraints.** Based on the computed RAS and FVA results, cell line–specific flux constraints were established using an approach adapted from [71]. For each reaction $j$ ($j = 1, \dots, R$) and cell line $c$ ($c = 1, \dots, C$), the flux bounds were computed as follows:

$$U_j^c = v_j^{max} \times \frac{RAS_j^c}{\max_c RAS_j^c} \tag{2}$$

$$L_j^c = v_j^{min} \times \frac{RAS_j^c}{\max_c RAS_j^c} \tag{3}$$

where $v_j^{max}$ and $v_j^{min}$ are the upper and lower bounds obtained from Flux Variability Analysis, $RAS_j^c$ denotes the Reaction Activity Score for reaction $j$ in cell line $c$, and $\max_c RAS_j^c$ is the maximum RAS value for reaction $j$ across all cell lines.

The lower bound for the biomass pseudo-reaction was set to 0.01 in CCLE-derived metabolic models to enforce minimal proliferation, since cancer cell lines inherently display a basal level of growth. Therefore, each cell line–specific model includes constraints derived from transcriptomic data and defined in Eqs 3 and 2.

## 4.4 Flux sampling

To explore the feasible flux space of each cell-specific metabolic model without assuming a predefined objective, we employed flux sampling. This method generates multiple flux distributions that satisfy steady-state and constraint-based conditions, offering a probabilistic view of metabolic activity beyond the scope of traditional objective-driven approaches like FBA.

In this study, we employed the Corner-based Sampling (CBS) algorithm to sample the vertices of the feasible flux space by iteratively optimizing randomly weighted objective functions [12,13]. Specifically, we used the implementation proposed by Galuzzi et al. [13] to maximize coverage of the feasible solution space. To this end, weights were assigned to reactions by sampling from a uniform distribution in the range [–1,1], and the optimization direction (maximization or minimization) was randomly selected at each iteration. To account for differences in reaction scales, each weight was normalized by the corresponding maximum flux value obtained from FVA. For each cell-specific model, we generated 10,000 flux samples.

## 4.5 Non-canonical TCA activity metrics

To quantify the activity of the non-canonical TCA cycle across cell lines, we defined two complementary metrics: *Cycle Propensity* and *Cycle Flux Intensity*. Both were computed independently for each cell-specific metabolic model, indexed by $c$.

*Cycle Propensity* captures the likelihood with which a cell line activates the non-canonical cycle. This was defined as the fraction of sampled flux distributions in which all three hallmark reactions—citrate export via SLC25A1, cleavage by ACLY, and malate regeneration via MDH1—exceed a minimal flux threshold ($\varepsilon = 10^{-6}$) and operate in the expected physiological direction.

$$\delta(f) = \begin{cases} 1 & \text{if } v_{\text{ACLY}} > \varepsilon \wedge v_{\text{SLC25A1}} > \varepsilon \wedge v_{\text{MDH1}} > \varepsilon \\ 0 & \text{otherwise} \end{cases} \tag{4}$$

Here, $\delta(f)$ is a binary indicator function assigning 1 to active flux samples and 0 otherwise; $v_{\text{ACLY}}, v_{\text{SLC25A1}}, v_{\text{MDH1}}$ denote the fluxes through the corresponding reactions in sample $f$.

$$\text{Propensity}_c = \frac{1}{|\mathcal{F}_c|} \sum_{f \in \mathcal{F}_c} \delta(f) \tag{5}$$

In this expression, $\mathcal{F}_c$ is the set of all flux samples generated for cell line $c$.

*Cycle Flux Intensity* measures the average operational strength of the cycle, conditional on activation. For each reaction $r \in R = \{\text{ACLY}, \text{SLC25A1}, \text{MDH1}\}$, we computed the mean flux across the subset of active samples. The final intensity value was defined as the minimum of these means, representing the rate-limiting step of the cycle:

$$\text{Flux Intensity}_c = \min_{r \in R} \left( \text{mean}_{f \in \mathcal{F}_c^{\text{active}}} v_r \right) \tag{6}$$

where $\mathcal{F}_c^{\text{active}} = \{f \in \mathcal{F}_c : \delta(f) = 1\}$. This formulation ensures that the metric reflects the bottleneck flux among the core reactions, thus offering a conservative estimate of cycle throughput in cell line $c$.

### 4.6 Machine learning framework

To identify genes whose expression is associated with non-canonical TCA cycle activity across different cancer cell lines, we developed a supervised regression pipeline to predict two activity metrics—*Cycle Propensity* and *Cycle Flux Intensity*. The input matrix *X* consisted of preprocessed gene expression data, while the output matrix *y* contained the two corresponding activity metrics for each cell line.

To capture both linear and non-linear associations between gene expression and metabolic phenotypes, we adopted two complementary feature selection strategies: *ElasticNet*, a linear model promoting variable selection through $\ell_1/\ell_2$ regularization, and *XGBoost feature importance*, a tree-based method capable of modeling complex, higher-order interactions among features.

**4.6.1 Data preprocessing.** For downstream machine learning analyses, the gene expression matrix (not processed with MAGIC imputation) was refined through a multi-step filtering procedure to enhance signal quality and model interpretability. Specifically, we applied four sequential filters. First, all pseudogenes were excluded based on Ensembl biotype annotations, as they are unlikely to exert functional regulatory effects. Second, genes encoding metabolic enzymes were systematically removed using curated protein class annotations from the Human Protein Atlas. Third, genes expressed at very low levels (below 0.5 TPM) in fewer than 10% of samples were excluded to avoid expression programs restricted to specific cell-line subtypes. Finally, genes with low cross-sample variability (bottom 10% of variance) were removed. After applying all filters, the feature set was reduced from 47,020 to 18,082 transcripts retained for downstream analyses.

**4.6.2 Co-expression module identification.** To mitigate the impact of highly correlated features during feature selection, we implemented a graph-based gene aggregation strategy prior to model training. Pairwise Pearson correlations were computed across all genes in the filtered expression matrix, and a signed mutual *k*-nearest neighbor graph ($k = 5$) was constructed using positive correlations ($r \geq 0.68$). Two genes were connected if each ranked the other among their top *k* positively correlated neighbors. Connected components in this mutual graph were then defined as clusters of highly co-expressed genes. This procedure yielded a total of 1506 multi-gene co-expression modules ($n \geq 2$), alongside 12949 singleton nodes, with the largest connected component comprising 365 genes.

**4.6.3 Data splitting, scaling, and module-wise dimensionality reduction.** The dataset was randomly shuffled before partitioning and divided into 72% for training, 18% for validation, and 10% for independent testing. We employed a nested cross-validation strategy comprising 10 outer folds and 5 inner folds. The inner loop was used for hyperparameter optimization, while the outer loop provided an unbiased evaluation of predictive performance and enabled the assessment of feature selection stability across folds.

Within each outer training fold, feature standardization was performed using the `StandardScaler`. The mean and standard deviation were computed exclusively on the training data and subsequently applied to the validation and test partitions.

Principal component analysis (PCA) was then applied to each co-expression module previously identified, using only the training data. For each module, the first principal component (PC1) was retained as a representative feature. The validation and test data were subsequently projected onto the PC1 loadings derived from the training set.

**4.6.4 MultiTask ElasticNet.** We applied a *MultiTask ElasticNet* regression model, a regularized linear approach that combines $\ell_1$ and $\ell_2$ penalties to jointly model associations between non-metabolic gene expression and both non-canonical TCA cycle metrics [72]. Since the two target variables were correlated, adopting a multitask learning strategy enabled the model to leverage their shared variance and to identify common transcriptional drivers underlying both metrics.

Model training and evaluation were performed within the nested cross-validation framework described above. Hyperparameters were optimized via Bayesian search using the Tree-structured Parzen Estimator (TPE) implemented in the `Hyperopt` library [73]. The search space included the overall regularization strength. $\ell_1/\ell_2$ mixing ratio and the inner-loop objective was to minimize the mean squared error across validation folds. Hyperparameter selection followed the

one–standard–error (1-SE) rule, favouring simpler and more strongly regularised configurations whose validation error lay within one standard error of the minimum. Model performance on training and test data across outer folds for the Multi-Task ElasticNet models is reported in S1 Fig. To further characterize the learning behavior of the model, we performed an ablation analysis using the optimized hyperparameters derived from the full training folds. For each of the ten outer folds, the model was retrained on progressively larger fractions of the available training data, ranging from 0.3 to 1.0 in increments of 0.1, with five random repetitions per fraction. The resulting $R^2$ and RMSE metrics were averaged across folds to generate the learning curves shown in S3 Fig.

For each outer fold, we ranked genes in descending order based on the absolute value of their regression coefficients and computed the cumulative sum of absolute weights. The optimal cutoff for feature retention was automatically determined using the `Kneedle` algorithm [74]. Genes contributing up to this threshold were retained as informative predictors. The fold-specific threshold values applied are reported in S3 Table.

**4.6.5 Multi-output XGBoost regression.** In parallel, we trained gradient boosting regression models using XGBoost [75] to jointly predict both non-canonical TCA cycle metrics from gene expression. Given the correlation between the two target variables, the model was configured to use multi-output trees (`multi_strategy = multi_output_tree`), allowing each tree to predict both outputs simultaneously and capture shared information across tasks.

Training and evaluation followed the same nested cross-validation and hyperparameter optimization procedure described for the ElasticNet models. The optimized parameters included the number of estimators, learning rate, maximum tree depth, minimum child weight, subsample ratio, column sampling rate, and both $\ell_1$ and $\ell_2$ regularization terms. Hyperparameters were selected using the 1-SE rule. Model performance across folds for the multi-output XGBoost models is reported in S2 Fig. The same ablation procedure described for the ElasticNet model was applied here to assess learning behavior (S4 Fig).

Feature importance was estimated using the "weight" method, which measures how frequently each feature was used for splitting across all trees. This was the only importance metric available in XGBoost for models trained with the multi-output tree configuration in the current software release (*ver 3.0.5*). For each outer fold, genes were ranked according to their "weight" importance values, and cumulative importance curves were used to identify the optimal cutoff for feature retention via the `Kneedle` algorithm [74]. The corresponding thresholds are reported in S4 Table.

**4.6.6 Stable feature selection.** To ensure that the identified predictors were not specific to a particular data split or training set composition, we quantified the stability of feature selection across outer cross-validation folds. This procedure aimed to identify genes that were consistently informative regardless of sample partitioning, thereby enhancing the robustness and reproducibility of the inferred associations.

For each predictive model (MultiTask ElasticNet and multi-output XGBoost), binary selection masks were generated to indicate whether a given gene was retained within each outer fold. For gene $i$, a stability score $s_i \in [0, 1]$ was computed as the proportion of outer folds in which the gene was selected:

$$s_i = \frac{1}{K} \sum_{k=1}^{K} m_i^{(k)} \tag{7}$$

where $K = 10$ is the number of outer folds and $m_i^{(k)} \in \{0, 1\}$ indicates whether gene $i$ was selected in fold $k$. Genes with $s_i \geq 0.7$ were considered stable, reflecting consistent selection across cross-validation folds.

**4.6.7 Gene-level interpretation via SHAP.** To quantify the local contribution of individual genes across cell lines, we computed SHAP values for both MultiTask ElasticNet and multi-output XGBoost models refitted on the subset of consistently selected genes identified across folds (no PCA applied to preserve single-gene interpretability). This approach ensured that interpretability remained focused on transcriptional predictors with demonstrated relevance, rather than being diluted by thousands of genes with negligible influence on model output, while also improving computational

efficiency. Moreover, since the predictive models were originally trained on co-expression modules, refitting them on the selected genes enabled SHAP to provide interpretable, cell line–specific attributions at single-gene resolution.

To ensure a fair and informative SHAP analysis, we first benchmarked both models across multiple feature panels derived exclusively from cross-fold–stable genes. The panels were: (i) the union panel, i.e., genes stably selected by either model (436 genes); (ii) the intersection panel, i.e., genes stably selected by both models (72); (iii) model-specific panels, i.e., ElasticNet-only (250) and XGBoost-only (114); and (iv) model-total panels, i.e., all stable genes from each model, ElasticNet-all (322) and XGBoost-all (186). The comparative performance across these panels (S11 Fig and S5 and S6 Tables) motivated the selection of the *union* panel for subsequent SHAP-based interpretation.

Training and evaluation followed the same nested cross-validation and hyperparameter optimization strategy used throughout the study. SHAP values were estimated using the `KernelExplainer`, with 400 background samples randomly drawn from the training data of each fold (out of 461 total) and were computed on the corresponding outer-test sets to obtain per–cell-line attributions. SHAP matrices from all outer folds were concatenated; gene-level importance was summarized as the mean absolute SHAP value across samples and folds, and the stability of these estimates was assessed across outer folds.

## 4.7 Protein–protein interaction network analysis

We constructed a protein–protein interaction (PPI) network using the STRING database *ver. 12.0* [76], using as interaction sources experimental evidence and curated database entries, with a minimum interaction confidence score of 0.4. Network clustering was performed using the Markov Clustering Algorithm with an inflation parameter of 1.5, resulting in 27 distinct clusters.

## 4.8 Over-representation analysis

Over-Representation Analysis was conducted on the final set of 436 selected genes. Enrichment was tested across three major gene set collections: Gene Ontology Biological Processes, KEGG, and MSigDB Hallmark gene sets.

The background gene set included 18,082 transcripts. Only terms containing fewer than 1,000 genes were retained for analysis. Enrichment significance was assessed using a hypergeometric test with Benjamini–Hochberg correction for multiple testing, and only terms with an FDR below 0.05 were considered significant.

ORA was carried out using the online tool `ShinyGO` *version 0.85.1* [77]. To reduce redundancy among enriched GO terms, results were post-processed using `REViGO` *ver. 1.8.2* [78]. Redundancy filtering was applied using the "Small (0.5)" setting for semantic similarity.

## 4.9 Gene dependency correlation analysis

We performed correlation analyses between both cycle activity metrics and genome-wide CERES gene effect scores from the DepMap project. To account for confounding, both the activity metrics and each gene dependency profile were residualized with respect to a common confounder matrix including a proliferation score, *TP53* mutation status, and histology fixed effects. By removing the variance explained by these covariates through ordinary least squares (OLS) regression, we obtained residualized values of the activity variable and each gene dependency profile that represent their components independent of proliferation, *TP53*, and histology effects. Spearman correlations were then computed between the residuals to quantify the remaining associations. We controlled the false discovery rate via Benjamini–Hochberg (*FDR* < 0.05). For prespecified Cit-Mal Cycle genes, no multitest adjustment was applied.

The proliferation score included in the confounder matrix was derived from cell cycle phase–specific transcriptional programs, as progression through the cell cycle is intrinsically linked to cellular proliferation. The S-phase and G2/M-phase scores were computed using the `score_genes` function from `Scanpy`, based on the human cell cycle marker gene

lists described by Tirosh et al. [79] . The proliferation score was then defined as the maximum between the S-phase and G2/M-phase scores.

## Supporting information

**S1 Fig. Fold-wise training and test prediction performance of the MultiTask ElasticNet model.** Boxplots showing the distribution of $R^2$ and RMSE values across the outer folds of nested cross-validation for both targets (*Cycle Propensity* and *Cycle Flux Intensity*) on training and test sets. Fold-wise test-set predictions are shown versus true values for *Cycle Propensity* and *Cycle Flux Intensity*. Across the outer folds of nested cross-validation, the model showed consistent test performance, attaining $\overline{R^2}_{Propensity} = 0.778 \pm 0.047$ and $\overline{R^2}_{Intensity} = 0.768 \pm 0.038$.
(TIFF)

**S2 Fig. Fold-wise training and test prediction performance of the multi-output XGBoost model.** Boxplots showing the distribution of $R^2$ and RMSE values across the outer folds of nested cross-validation for both targets (*Cycle Propensity* and *Cycle Flux Intensity*) on training and test sets. Fold-wise test-set predictions are shown versus true values for *Cycle Propensity* and *Cycle Flux Intensity*. Across the outer folds of nested cross-validation, the model showed consistent test performance, attaining $\overline{R^2}_{Propensity} = 0.813 \pm 0.047$ and $\overline{R^2}_{Intensity} = 0.806 \pm 0.024$.
(TIFF)

**S3 Fig. Learning curve of MultiTask ElasticNet models.** Learning curves showing model performance across increasing fractions of training data (0.3–1.0, step 0.1; five repetitions per fold). Each panel reports mean $\pm$ s.d. across ten outer folds of nested cross-validation. For both targets, predictive performance improves consistently as the amount of training data increases, with a train–test gap ($\Delta R^2 \approx 0.07$–0.09) that gradually narrows at higher data fractions. This pattern indicates that the model continues to benefit from additional training data while maintaining a relatively small and decreasing train–test discrepancy across folds.
(TIFF)

**S4 Fig. Learning curve of multi-output XGBoost models.** Learning curves showing model performance across increasing fractions of training data (0.3–1.0, step 0.1; five repetitions per fold). Each panel reports mean $\pm$ s.d. across ten outer folds of nested cross-validation. For both targets, test performance increases steadily with larger training fractions, while a moderate train–test gap ($\Delta R^2 \approx 0.12$) persists across data fractions. The curves correspond to the final models obtained after hyperparameter tuning under a 1-SE selection strategy with constrained, strongly regularized search spaces. Despite this residual gap, test performance is generally stable and reproducible across folds.
(TIFF)

**S5 Fig. Full PPI network of the 361 genes mapped to STRING.** Edges represent high-confidence interactions (score $\geq 0.4$) from experimental and curated database sources. Nodes corresponding to the top-20 SHAP-predictive genes (from both models and targets) are highlighted in blue, while genes recurring among the top predictors across all four SHAP analyses (*PLEKHA6* and *SELENBP1*) are shown in red. Of the 436 predictive genes, 361 formed 229 interactions, significantly above random expectation ($p < 10^{-16}$).
(TIFF)

**S6 Fig. Functional clustering of predictive genes within the STRING PPI network.** The network was clustered into three main modules (cluster size $\geq 10$ genes): a red cluster (21 genes) enriched for *extracellular matrix–receptor interaction*; a green cluster (18 genes) enriched for the *RHO GTPase cycle*; and a brown cluster (10 genes) enriched for *RUNX3–YAP1–mediated transcription* and *transcriptional regulation of granulopoiesis*.
(TIFF)

**S7 Fig. Mean absolute SHAP values and stability of gene importance across outer folds for the MultiTask ElasticNet model.** Bar plots display the mean absolute SHAP values (|SHAP|) with standard deviation across outer cross-validation folds, reflecting the average contribution and stability of each gene to model predictions. Panel (A) reports the top 20 predictive genes for *Cycle Propensity*, and panel (B) the top 20 predictive genes for *Cycle Flux Intensity*.
(TIFF)

**S8 Fig. Mean absolute SHAP values and stability of gene importance across outer folds for the multi-output XGBoost model.** Bar plots display the mean absolute SHAP values (|SHAP|) with standard deviation across outer cross-validation folds, reflecting the average contribution and stability of each gene to model predictions. Panel (A) reports the top 20 predictive genes for *Cycle Propensity*, and panel (B) the top 20 predictive genes for *Cycle Flux Intensity*.
(TIFF)

**S9 Fig. Significant gene dependencies correlated with *Cycle Propensity*.** Scatter plots showing the relationship between residualized *Cycle Propensity* and residualized gene–effect scores across cancer cell lines for significant associations (FDR <0.05). Correlations were computed using Spearman's rank correlation.
(TIFF)

**S10 Fig. Flux-based propensity of canonical and non-canonical TCA cycles in the Recon3D model.** Boxplots showing the estimated propensity of the canonical and non-canonical TCA cycles across 513 cancer cell lines, computed from 1,000 flux samples per line using the Recon3D model. A sample was classified as *active* for the canonical TCA cycle when all reactions composing the mitochondrial loop carried nonzero fluxes in the clockwise direction. The same sampling settings and parameters used for ENGRO2.2 core models were applied here. Both cycles showed low activation frequency.
(TIFF)

**S11 Fig. Performance of feature panels across models and targets.** Boxplots showing the distribution of $R^2$ and RMSE values across outer folds of nested cross-validation for six feature panels: *Union*, *Intersection*, *XGB-only*, *EN-only*, *XGB-all*, and *EN-all*. Each point represents the performance in one outer fold. Panel (A) reports results for the *MultiTask ElasticNet* model and panel (B) for the *multi-output XGBoost* model. Statistical comparisons between panels are reported in Supplementary Tables S4 and S5.
(TIFF)

**S1 Table. List of selected genes.** Comprehensive list of the 436 selected genes (Ensembl gene identifiers) that were retained in the final predictive gene panel used for downstream modeling and interpretation.
(XLSX)

**S2 Table. Enrichment statistics for the 436 selected genes.** Enrichment statistics for all terms significantly enriched among the 436 selected genes. For each term, the table reports the gene set, number of overlapping genes, fold enrichment, and Benjamini–Hochberg adjusted FDR.
(XLSX)

**S3 Table. Kneedle thresholds across outer folds for the MultiTask ElasticNet model.** Table reporting the Kneedle threshold values estimated across the ten outer folds of nested cross-validation for the MultiTask ElasticNet model, summarizing the cutoffs used to select stable predictive genes based on mean absolute SHAP values.
(XLSX)

**S4 Table. Kneedle thresholds across outer folds for the multi-output XGBoost model.** Table reporting the Kneedle threshold values estimated across the ten outer folds of nested cross-validation for the multi-output XGBoost model, summarizing the cutoffs used to select stable predictive genes based on mean absolute SHAP values.
(XLSX)

**S5 Table. Pairwise Wilcoxon tests including only comparisons with the *Union* panel for the MultiTask Elastic-Net model.** Pairwise Wilcoxon signed-rank tests comparing the *Union* feature panel with the *Intersection*, *EN-only*, *EN-all*, *XGB-all*, and *XGB-only* panels for both non-canonical TCA cycle metrics. For each comparison, the table reports the median performance difference Δ (Panel A – Panel B), the *p*-value, FDR-corrected *q*-value, and the winning panel for both $R^2$ and RMSE. For $R^2$, Δ > 0 favors Panel A; for RMSE, Δ < 0 favors Panel A. (XLSX)

**S6 Table. Pairwise Wilcoxon tests including only comparisons with the *Union* panel for the multi-output XGBoost model.** Pairwise Wilcoxon signed-rank tests comparing the *Union* feature panel with the *Intersection*, *EN-only*, *EN-all*, *XGB-all*, and *XGB-only* panels for both non-canonical TCA cycle metrics under the multi-output XGBoost model. For each comparison, the table reports the median performance difference Δ (Panel A – Panel B), the *p*-value, FDR-corrected *q*-value, and the winning panel for both $R^2$ and RMSE. For $R^2$, Δ > 0 favors Panel A; for RMSE, Δ < 0 favors Panel A. (XLSX)

**S1 Data. SHAP values for each gene and cell line in ElasticNet and XGBoost models.** File containing SHAP values for *Cycle Propensity* and *Cycle Flux Intensity* for each cancer cell line. The workbook includes four sheets: `xgb_propensity` and `xgb_intensity` report SHAP values from the multi-output XGBoost model, while `en_propensity` and `en_intensity` report SHAP values from the Multi-Task ElasticNet model. In each sheet, rows correspond to individual cancer cell lines from the CCLE panel, and columns correspond to model input genes. Cell entries represent the SHAP value for a given gene, cell line, and fold, quantifying the local contribution of that gene to the predicted cycle-activity metric. (XLSX)

## Author contributions

**Conceptualization:** Lihao Lin, Marco Vanoni, Lilia Alberghina, Chiara Damiani.

**Funding acquisition:** Marco Vanoni, Lilia Alberghina, Chiara Damiani.

**Investigation:** Lihao Lin, Francesco Lapi.

**Methodology:** Lihao Lin, Francesco Lapi, Bruno Giovanni Galuzzi, Chiara Damiani.

**Software:** Lihao Lin, Francesco Lapi, Bruno Giovanni Galuzzi.

**Supervision:** Chiara Damiani.

**Visualization:** Lihao Lin.

**Writing – original draft:** Lihao Lin.

**Writing – review & editing:** Lihao Lin, Marco Vanoni, Chiara Damiani.

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
