## [Decision Letter · Decision Letter 0]

23 Sep 2025

PCOMPBIOL-D-25-01534

Mechanistically Informed Machine Learning Links Non-Canonical TCA Cycle Activity to Warburg Metabolism and Hallmarks of Malignancy

PLOS Computational Biology

Dear Dr. Damiani,

Thank you for submitting your manuscript to PLOS Computational Biology. After careful consideration, we feel that it has merit but does not fully meet PLOS Computational Biology's publication criteria as it currently stands. Therefore, we invite you to submit a revised version of the manuscript that addresses the points raised during the review process.

Please submit your revised manuscript within 60 days Nov 23 2025 11:59PM. If you will need more time than this to complete your revisions, please reply to this message or contact the journal office at ploscompbiol@plos.org. Please include the following items when submitting your revised manuscript:

We look forward to receiving your revised manuscript.

Kind regards,

Julio R. Banga, Ph.D.

Academic Editor

PLOS Computational Biology

Pedro Mendes

Section Editor

PLOS Computational Biology

**Journal Requirements:**

**Reviewers' comments:**

Reviewer's Responses to Questions

**Comments to the Authors:**

Reviewer #1: Summary

The submitted manuscript describes a study of the newly discovered Non-canonical TCA Cycle (Arnold Cycle) by a combination of constrained-based metabolic modeling and machine learning to identify transcriptomic signatures and marker genes of activity. Since the Arnold Cycle was described only very recently, the authors contribution is novel and interesting for a broader interdisciplinary audience and aims a generating testable hypotheses via in silico approaches in a classical systems biology cycle.

The manuscript is well written, results are well depicted with illustrations and reproducibility is ensured via a open-source repository including code for in silico approaches. Beside these strong points, I see one critical point and additional major as well as minor points (outlined below). The critical point of the study, in my view, is the fragility of the two-step approach without external or experimental validation of the first step where a newly developed metabolic is leveraged to predict pathway activity. As the authors themselves, this is crucial for the second step to predict other marker genes and overall results look fragile towards modelling choices and decisions in the first step. Hence, I suggest that authors outline or even apply a strategy to externally validate the first step gaining confidence and reliability of overall findings.

Critical

Study builds on two steps with predictions without external/experimental validation of the modeling and method. The Authors in the last paragraph acknowledge that metabolic modeling in the first step is crucial for analysis but do no outline or apply a strategy to validate their approach/results before applying ML in the second step. Further, they state that denoising of the transcriptome was necessary which additionally shows that there is a degree of fragility in the two-step approach.

Main

- The benefit of Arnold Cycle unclear: Is it Speed? Cost-Efficiency? Oxygen limit/Warburg … -> Please, address this more in Introduction and Discussion in the sense of “nothing makes sense except in the light of evolution” there must be hypotheses about the necessity of this metabolic route.

- “we trained regression models to predict both Cycle Propensity and Cycle Flux Intensity” -> In parts of the manuscript it is ambiguous that in fact fully independent ML-models were trained for each target variable. Please, make sure to clearly distinguish from Multi-Task Learning/Transfer-Learning approaches where a simultaneous optimization is pursuit. Perhaps, discuss why such approaches have not been used.

- Highly correlated features (genes) can severely hamper feature importance analysis with different sensitivity towards this issue across the used methods. Hence, I would suggest aggregating gene clusters of very high correlated genes before training models and performing importance analysis to increase robustness.

- I do not understand the choice of XGBoost in addition to the prior used RandomForests. Please, motivate more why XGBoost + SHAP are used for the final models and which benefit they have over RandomForest + feature importance.

- Details on train-test-valid splits. The authors perform a cross-validation strategy and I appreciate that robustness is checked via inner and outer-folds. However, information on how the data is split into training, valid and a final test set is scarce. Further, performance metrics should be reported also on the training set to see the degree of over/underfitting. Lastly, I suggest considering a splitting strategy on the cell line level by reserving e.g. 10% of the >500 cell lines as test-set to check the generalizability of the model.

Minor

- Please, provide line-numbers in the manuscript PDF. This makes giving feedback easier.

- Give some crucial method details in result part to understand approach without going to Method section or change overall order (Method section before Result)

-> Examples: (1) what are main and necessity updates of ENGRO2 -> 2.2? Clearly state that this model is new and not tested in other studies. (2) Constrained-based modeling approach: sampling approach, (non)-objective function. The metabolic modeling approach and objective remains unclear without fully reading the method section.

- “This inverse relationship suggests that a higher likelihood of Arnold Cycle engagement does not necessarily correspond to increased metabolic throughput.” -> I suggest rephrasing. “Not necessarily” implies more a neutral and not a strong inverse correlation.

- You may discuss more the interrelation of the Arnold Cycle, Angiogenesis and Warburg. I guess they are connected via low oxygen requirements.

- I understand the priority to relate DepMap with Cycle Propensity. But relating DepMap with Flux Intensity may also provide some insights and could be included in a supplement.

- Authors make the relation of citrate-to-alpha-KG with epigenetic regulation and cell fate decision. To which degree is this feasible based on the used metabolic model? Is this aspect even captured by the constrained-based modeling approach?

- “In particular, it yielded higher silhouette scores when clustering cell lines based on simulated flux distributions, and also led to significantly better alignment with transcriptome-based clusters defined by metabolic gene expression.” -> Do cell lines really need to form good clusters in the flux distribution space similarly to the transcriptome space? I suggest justifying more this assumption and to include the results of this prior analysis in the supplemental material in order to understand this.

- “In line with this, preliminary flux sampling experiments using the Recon3D [46] model revealed that both canonical and non-canonical TCA cycles were poorly represented. These results support the use of a core model for our analysis.” -> Similar as above, Authors refer to a prior analysis which is not included in the manuscript or a electronic supplement. I suggest including all of those to foster transparency.

- Why were weights for objective function sampled with negative and positive values with switching between maximization and minimization? Why not only maximization?

- The overlap between ElasticNet and RandomForest discovered genes (6) is surprisingly little. Filtering and aggregating highly correlated genes should improve this. I suggest having a closer look at those 6 genes and their role.

- When applying SHAP for estimating feature importance, the definition of a reference/background data set can be crucial. Please, give information and justification on what is used here. A self-defined set or a randomized set?

Reviewer #2: This is an interesting paper in which the authors use a combination of mechanistic modelling and machine learning to investigate the extent to which the non-canonical TCA cycle is active in approximately 500 cancer cell lines and show how its activity correlates with enrichment in genes associated with the hallmarks of cancer (eg angiogenesis, invasion, stemness). Excitingly, these findings provide new insights into how cancer cells can adapt their metabolism as they become more pathological and could serve as a basis for new targets for treating cancer.

The paper is well-written and logically structured, with the results clearly stated and developed in a logical manner. The methods used are outside my expertise so I am unable to comment on them (and hope that the other reviewers are able to comment on the methodology). From what I understand, the methods used seem to be fairly straight forward, but they are appropriate for the questions of interest and build on the authors’ earlier work.

Reviewer #3: Review: Lihao et al., investigated how cancer cells leverage metabolic circuits to support phenotypic plasticity and progression, with a particular focus on the non-canonical citrate metabolism. Recognizing the difficulty of inferring compartment-spanning fluxes from existing datasets, they develop a mechanistically informed machine learning framework that integrates constraint-based metabolic modeling with supervised learning. Using non-canonical citrate metabolic activity metrics derived from over 500 cancer cell lines, they identify transcriptional programs predictive of cell cycle activity. Their approach not only establishes robust transcriptional correlates of citrate metabolic flux but also uncovers broader links between this cycle and metabolic reprogramming, providing new insights into how metabolic flexibility underpins aggressive cancer phenotypes.

1- The manuscript designates the metabolic circuit as the Arnold Cycle. However, I note that the original foundational study on this pathway listed two co–first authors, both of whom should receive equivalent recognition. As such, if the authors wish to propose a new phenotype or formalize a naming convention, the terminology should reflect joint credit (e.g., Arnold–Jackson or Arnold–Jackson–Finely). Alternatively, a more neutral and descriptive designation, such as non-canonical citric metabolism, may be preferable to avoid ambiguity or inequity in attribution.

2- Figure 1b comes before Figure 1c in the text. The order should be fixed either in the figure or in the text.

3- is claimed that:” From these samples, we derived two complementary metrics: Cycle Propensity, defined as the fraction of sampled states in which the three core reactions (SLC25A1, ACLY, MDH1) of the Arnold Cycle are simultaneously active; and Cycle Flux Intensity, measuring the average flux through the cycle’s bottleneck reaction in active configurations (Fig. 1b).” However, Figure 2 doesn’t show this; and the two concepts needs much more clarifications.

4- Cancer is separated to stages and subtypes and any of the cell lines are assigned to types or subtypes of cancer such as MDA-mb-231 as triple negative metastatic breast cancer with high Warburg phenotype features or MCF7 as non-metastatic hormone positive cell types. These should be tested in the model comparing for example 231 cells as high Warburg phenotype compared to MCF7 as low Warburg phenotype and normal such as MCF10A. These can be also tested in section 2.2 for Warburg phenotype and oxygen consumption.

5- DepMap dependency correlations

6- For DepMap dependency correlation analysis, they correlate Propensity with CERES scores genome-wide and highlight WNT/KRAS/PI3K/MAPK patterns. It is critical to multiple-testing control and confounders (lineage, proliferation rate, TP53 status).

7- The manuscript remains purely in silico. Even a lightweight orthogonal validation would greatly strengthen the claims:

a) Select 2–3 “high-propensity/low-intensity” lines and 2–3 “low-propensity/high-intensity” lines; measure OCR/ECAR and lactate secretion (Seahorse/biochemical assays) to verify predicted trends;

b) Test sensitivity to ACLY (SB-204990, Bempedoic-acid CoA analog) and SLC25A1 (CTPi), plus IDH1 inhibition (AG-120 in IDH1-WT as control). Show differential viability/flux adaptation consistent with your dependency analyses;

c) Perform limited ¹³C-glucose/¹³C-citrate tracing to confirm cytosolic citrate routing via ACO1→IDH1 vs ACLY.

Minor Comments

Throughout: fix small typos (“bas been”, “characterizate”, “cine–specific”, “theWarburg”, “ACLY, promotes”), and a repeated “p = 5.50 < 10−91 / 6.24 < 10−110” formatting, should be “p < 5.5×10⁻⁹¹”, etc.

Figure 1A/B: add units for fluxes (arbitrary units are fine but state “normalized steady-state flux”).

Figure 2A: add sample sizes per lineage; consider rug plots/marginal densities for each axis.

Figure 3B: report odds ratios and top term counts; consider reducing redundancy with parent-child trimming.

SHAP plots: add fold-wise variation bars or stability heatmaps; ensure no metabolic genes slipped into the non-metabolic feature set (list in Supp. Table S1).

**Have the authors made all data and (if applicable) computational code underlying the findings in their manuscript fully available?**

Reviewer #1: Yes

Reviewer #2: Yes

Reviewer #3: **No: **

PLOS authors have the option to publish the peer review history of their article (what does this mean?). If published, this will include your full peer review and any attached files.

Reviewer #1: **Yes: **Jan Ewald

Reviewer #2: No

Reviewer #3: **Yes: **mehdi damaghi

**Figure resubmission:**
---

## [Decision Letter · Decision Letter 1]

1 Dec 2025

Dear Prof. Damiani,

We are pleased to inform you that your manuscript 'Mechanistically Informed Machine Learning Links Non-Canonical TCA Cycle Activity to Warburg Metabolism and Hallmarks of Malignancy' has been provisionally accepted for publication in PLOS Computational Biology.

Best regards,

Julio R. Banga, Ph.D.

Academic Editor

PLOS Computational Biology

Pedro Mendes

Section Editor

PLOS Computational Biology

Please check the suggestions of reviewer #1.

Reviewer's Responses to Questions

**Comments to the Authors:**

Reviewer #1: The authors of the manuscript have done a tremendous job in improving their study and precisely addressed the points raised by all reviewers. I really appreciate their efforts which led to a vastly improved manuscript and robust findings.

Hence, I do not have any new major points and overall suggest an acceptance of this valuable research. I have only two minor things after going through the revised manuscript:

- Section 2.5 and partly others look and read difficult with all the information in brackets about FDR/FE values. In my opinion it is enough that you state that they are all significant at FDR <0.05 and not provide all values which can be part of an electronic supplement.

- Check any length restrictions by the journal. Manuscript is now pretty long after the revision (naturally). You may consider shortening if deemed necessary by the editorial team.

Reviewer #3: The authors addressed all of my comments and questions.

**Have the authors made all data and (if applicable) computational code underlying the findings in their manuscript fully available?**

Reviewer #1: Yes

Reviewer #3: Yes

PLOS authors have the option to publish the peer review history of their article (what does this mean?). If published, this will include your full peer review and any attached files.

Reviewer #1: **Yes: **Jan Ewald

Reviewer #3: **Yes: **Mehdi Damaghi

---

## [Editor Report · Acceptance letter]

PCOMPBIOL-D-25-01534R1

Mechanistically Informed Machine Learning Links Non-Canonical TCA Cycle Activity to Warburg Metabolism and Hallmarks of Malignancy

Dear Dr Damiani,

I am pleased to inform you that your manuscript has been formally accepted for publication in PLOS Computational Biology. Your manuscript is now with our production department and you will be notified of the publication date in due course.

With kind regards,

Anita Estes
